# Identifying putative ventilation-perfusion distributions in COVID-19 pneumonia

**Haopeng Xu**[1], **Nayia Petousi**[1], **Peter A. Robbins**[2]*

**1** Nuffield Department of Medicine, University of Oxford, Oxford, United Kingdom, **2** Department of Physiology, Anatomy and Genetics, University of Oxford, Oxford, United Kingdom

* peter.robbins@dpag.ox.ac.uk

**Data Availability Statement:** All relevant data are within the paper.

**Funding:** HX, China Scholarship Council (CSC)-Chinese Academy of Medical Sciences (CAMS)

## Abstract

Busana *et al*. (doi.org/10.1152/japplphysiol.00871.2020) published 5 patients with COVID-19 in whom the fraction of non-aerated lung tissue had been quantified by computed tomography. They assumed that shunt flow fraction was proportional to the non-aerated lung fraction, and, by randomly generating $10^6$ different bimodal distributions for the ventilation-perfusion ($\dot{V}/\dot{Q}$) ratios in the lung, specified as sets of paired values $\{\dot{V}i, \dot{Q}i\}$, sought to identify as solutions those that generated the observed arterial partial pressures of $CO_2$ and $O_2$ ($Pa_{CO2}$ and $Pa_{O2}$). Our study sought to develop a direct method of calculation to replace the approach of randomly generating different distributions, and so provide more accurate solutions that were within the measurement error of the blood-gas data. For the one patient in whom Busana *et al*. did not find solutions, we demonstrated that the assumed shunt flow fraction led to a non-shunt blood flow that was too low to support the required gas exchange. For the other four patients, we found precise solutions (prediction error $< 1\times10^{-3}$ mmHg for both $Pa_{CO2}$ and $Pa_{O2}$), with distributions qualitatively similar to those of Busana *et al*. These distributions were extremely wide and unlikely to be physically realisable, because they predict the maintenance of very large concentration gradients in regions of the lung where convection is slow. We consider that these wide distributions arise because the assumed value for shunt flow is too low in these patients, and we discuss possible reasons why the assumption relating to shunt flow fraction may break down in COVID-19 pneumonia.

## Introduction

COVID-19 pneumonia is unusual in the severity of the hypoxaemia relative to the degree of atelectasis or consolidation observed in the lung [1]. To explore gas exchange in COVID-19 pneumonia further, *Busana et al.* [2] enrolled 5 patients with the disease who were undergoing mechanical ventilation in an intensive care unit, who had a pulmonary artery catheter in place for clinical reasons, and for whom there was a complete set of data for gas exchange, haemodynamics and lung mechanics together with a near-contemporaneous chest CT scan.

Busana *et al.* sought to interpret their data in terms of the associated ventilation-perfusion ($\dot{V}/\dot{Q}$) distribution in the lung. In order to do this, it was necessary to assign a shunt fraction (the fraction of the cardiac output that passes through the lungs without coming into contact

Oxford Institute (COI), https://www.camsoxford.ox.ac.uk/. NP, Respiratory theme, the National Institute for Health Research (NIHR) Oxford Biomedical Research Centre (BRC) http://oxfordbrc.nihr.ac.uk/ PAR, Respiratory theme, the National Institute for Health Research (NIHR) Oxford Biomedical Research Centre (BRC) http://oxfordbrc.nihr.ac.uk/ The views expressed are those of the authors and not necessarily those of the National Health Service (NHS), the NIHR, or the Department of Health. The funders had no role in study design, data collection and analysis, decision to publish, or preparation of the manuscript.

**Competing interests:** The authors have declared that no competing interests exist.

with any fresh gas) for each patient. Busana *et al.* assigned this shunt fraction as equivalent to the fraction of non-aerated lung tissue observed with quantitative computed tomography for each of five patients with severe COVID-19 pneumonia. This assumption is important and we refer to it henceforth as the 'shunt fraction assumption'. The remainder of the cardiac output perfused the aerated lung tissue, and with this Busana *et al.* sought to find solutions to their problem in form of ventilation-perfusion ($\dot{V}/\dot{Q}$) distributions that would reproduce the arterial partial pressures of $CO_2$ and $O_2$ ($Pa_{CO2}$ and $Pa_{O2}$, respectively) observed in the patients.

Busana *et al.*'s model of gas exchange had 498 compartments that were both perfused and ventilated, with values for $\dot{V}/\dot{Q}$ ranging from between ~$10^{-2}$ to ~$10^2$. Any particular $\dot{V}/\dot{Q}$ distribution is then specified as a set of paired values for ventilation and perfusion {$\dot{V}i$, $\dot{Q}i$}, where i is the index for the compartment. Each compartment has a single value for $Pa_{CO2}$ and $Pa_{O2}$ associated with it that is determined by its $\dot{V}/\dot{Q}$ ratio, and the compartments taken together represent the variation in $Pa_{CO2}$ and $Pa_{O2}$ across the lungs. The $Pa_{CO2}$ and $Pa_{O2}$ for the whole lung can be calculated by mixing together all of the blood leaving the compartments. Thus the putative or candidate $\dot{V}/\dot{Q}$ distributions for any particular patient are those that, when the blood is combined from all the compartments, result in $Pa_{CO2}$ and $Pa_{O2}$ values that match those of the patient.

Busana *et al.* used an approach of randomly generating many different $\dot{V}/\dot{Q}$ distributions to identify candidate distributions that may approximate the underlying distribution within each patient. For each patient, $10^6$ bimodal distributions were randomly generated, based on five underlying parameters. For each distribution, an associated $Pa_{CO2}$ and $Pa_{O2}$ value were calculated. Distributions were considered as potentially acceptable solutions for the underlying $\dot{V}/\dot{Q}$ distribution of the patient if the resultant $Pa_{CO2}$ and $Pa_{O2}$ values were within 10% of the measured values.

Busana *et al.* found potential solutions for four out of the five patients. The predicted values for $Pa_{CO2}$ and $Pa_{O2}$ from all the potential solutions for the $\dot{V}/\dot{Q}$ distribution for each patient together with the patient's actual $Pa_{CO2}$ and $Pa_{O2}$ are illustrated in the figure numbered six in their paper of their paper. For no patient did their approach generate any candidate $\dot{V}/\dot{Q}$ distribution where the model value for either $Pa_{CO2}$ or $Pa_{O2}$ fell within 1 mmHg of the measured value, which is a reasonable estimate for the error associated with a blood-gas measurement. Furthermore, for no patient did the cloud of potential solution points surround the patient's measured value. Instead the cloud was always located away from the true value in the direction of higher $Pa_{CO2}$ and higher $Pa_{O2}$. It is not clear whether these features arise because of the limited accuracy of the random simulation approach or whether there are other, more fundamental factors involved.

The purpose of the present study was to improve the accuracy with which candidate $\dot{V}/\dot{Q}$ distributions could be identified. In particular we sought to replace the random simulation approach by developing a method that would allow the direct calculation of the parameter set(s) for a $\dot{V}/\dot{Q}$ distribution that would reproduce a given patient's $Pa_{CO2}$ and $Pa_{O2}$. Three different types of $\dot{V}/\dot{Q}$ distribution were explored in increasing order of complexity: 1) the well-known three-compartment model; 2) a model with one low and one high $\dot{V}/\dot{Q}$ compartment; and 3) a continuous $\dot{V}/\dot{Q}$ distribution based on the beta distribution.

## Methods

### Overview

In this study, a set of paired values for ventilation and perfusion, {$\dot{V}i$, $\dot{Q}i$}, is referred to as a $\dot{V}/\dot{Q}$ distribution and reflects the ventilation and perfusion going to different regions (indexed

by i) in the lung. A particular choice of the number of elements within $\{\dot{V}i, \dot{Q}i\}$, or a particular choice of the distribution of the ratios $\dot{V}i/\dot{Q}i$ within the set, is referred to as a compartmental model. Also referred to as a model within this study, is a set of equations, based on mass balance and blood gas chemistry, that reflect the processes of gas exchange within the lung. When supplied with the particular inspiratory and mixed venous blood gas compositions pertaining to a patient, this model maps $\{\dot{V}i, \dot{Q}i\}$ to an associated arterial blood gas composition. Solutions are sets $\{\dot{V}i, \dot{Q}i\}$ for which this calculated arterial blood gas composition matches the measured arterial blood gas composition of the patient to within experimental error. In this study, the problem is to find solutions that are also consistent with additional constraints relating to the patient, in particular the value for the cardiac output (the sum of all the elements, $\dot{Q}i$), the shunt flow (the value for $\dot{Q}i$, for which $\dot{V}i = 0$) and the measured rate of oxygen uptake.

The methods are split into three main sections. The first section is a statement of the governing equations for compartmental models of gas exchange. The second section develops a set of useful functions, based on the governing equations of the model, that map between $\dot{V}/\dot{Q}$ ratios, respiratory quotient (R) values and compartmental partial pressures for $CO_2$ and $O_2$ ($P_{CO_2}$ and $P_{O_2}$, respectively). The third section develops the methodology to calculate parameter sets that will reproduce the $Pa_{CO_2}$ and $Pa_{O_2}$ values for each of three types of compartmental model under consideration. A flow chart to illustrate the overall process is given in Fig 1. The accuracy we sought was for the model values for $Pa_{CO_2}$ and $Pa_{O_2}$ to be within 1 mmHg of the patient's measured values, although in practice the errors were less that $1\times10^{-2}$ mmHg or even $1\times10^{-3}$ mmHg.

All calculations were performed in Matlab version 9.9.0. Where numerical solutions were required, Matlab's inbuilt solvers fsolve and fminbnd were used.

## Governing equations

The lung is heterogenous, with different partial pressures for alveolar $CO_2$ and $O_2$ occurring in different locations. One way of modelling this is to consider the lung as if it were constructed from a set of compartments, with each individual compartment having its own unique values for alveolar $P_{CO_2}$ and $P_{O_2}$. The governing equations for the model of compartmental gas exchange arise from the conservation of mass and are essentially those from the classical studies surrounding $\dot{V}/\dot{Q}$ developed by Fenn, Rahn and Otis [3, 4] and by Riley and Cournand [5, 6].

In any given compartment, the rates at which $CO_2$, $O_2$ and $N_2$ enter or leave the compartment from the blood have to equal to the rates that they enter or leave from the gas phase. This yields the following equations:

$$\dot{V}_{CO2} = \dot{Q}(C\bar{v}_{CO2} - C_{CO2}) = \dot{V}EP_{CO2}/PB - \dot{V}IPI_{CO2}/PB, \tag{1}$$

$$\dot{V}_{O2} = \dot{Q}(C_{O2} - C\bar{v}_{O2}) = \dot{V}IPI_{O2}/PB - \dot{V}EP_{O2}/PB, \text{ and} \tag{2}$$

$$\dot{V}_{N2} = 0 = \dot{V}IPI_{N2}/PB - \dot{V}EP_{N2}/PB. \tag{3}$$

where $\dot{V}_{CO2}$, $\dot{V}_{O2}$ and $\dot{V}_{N2}$ are the compartmental rates of gas production or consumption for $CO_2$, $O_2$ and $N_2$, respectively; $\dot{Q}$ is perfusion of the compartment; $C\bar{v}_{CO2}$ and $C\bar{v}_{O2}$ are mixed venous contents of $CO_2$ and $O_2$, respectively; $C_{CO2}$ and $C_{O2}$ are the compartmental end-capillary blood contents for $CO_2$ and $O_2$, respectively; $P_{CO2}$ and $P_{O2}$ are the compartmental partial pressures for $CO_2$ and $O_2$, respectively; $PI_{CO2}$ and $PI_{O2}$ are inspired partial pressures for $CO_2$

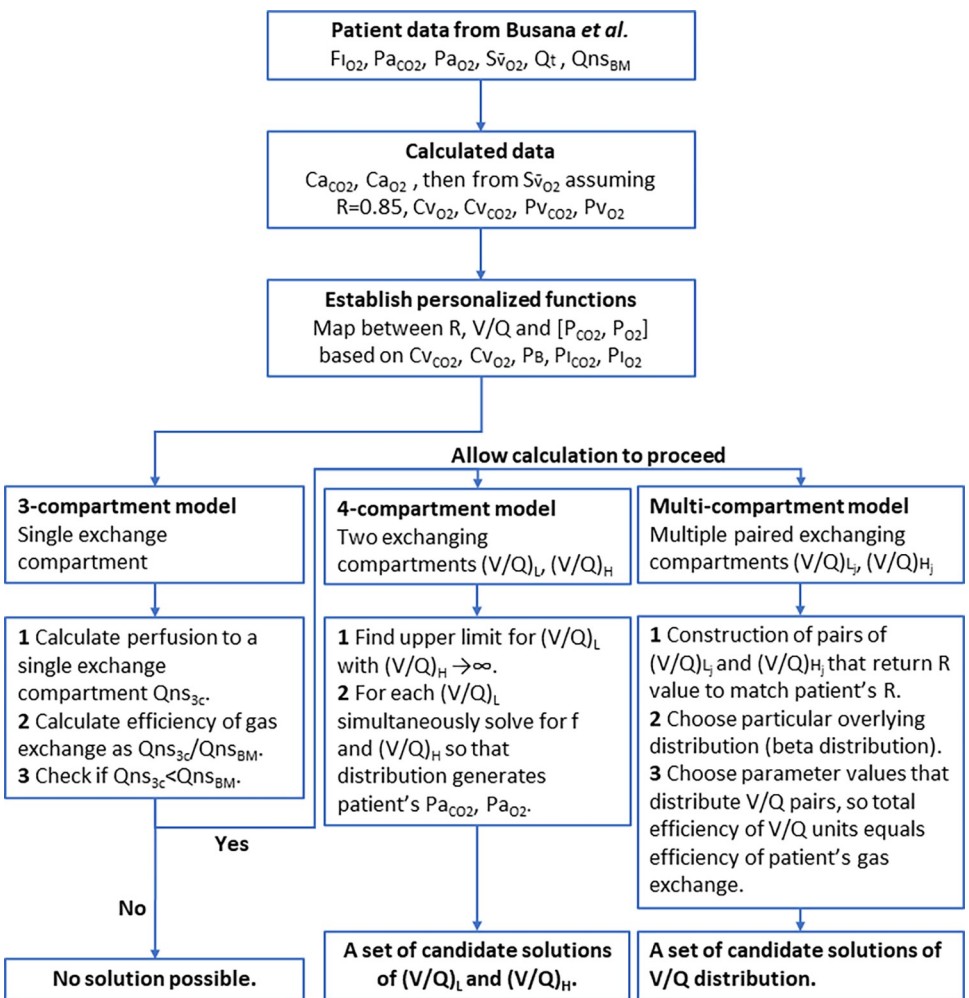

**Fig 1. Flow chart of the overall process.** The flow chart illustrates the source of data, the analytic approach taken, and the expected results. $FI_{O2}$, inspired $O_2$ fraction; $Pa_{CO2}$ & $Pa_{O2}$, arterial partial pressures for $CO_2$ and $O_2$, respectively; $S\bar{v}_{O2}$, $O_2$ saturation of mixed venous blood; $\dot{Q}t$, total cardiac output; $\dot{Q}ns_{BM}$, non-shunt flow from Busana et al.; $Ca_{CO2}$ & $Ca_{O2}$, arterial gas contents for $CO_2$ and $O_2$, respectively; R, respiratory quotient; $C\bar{v}_{O2}$ & $C\bar{v}_{CO2}$, mixed venous gas contents for $O_2$ and $CO_2$, respectively; $P\bar{v}_{CO2}$ & $P\bar{v}_{O2}$, mixed venous partial pressures for $CO_2$ and $O_2$, respectively; $P_B$, barometric pressure; $PI_{CO2}$ & $PI_{O2}$, inspired partial pressures for $CO_2$ and $O_2$, respectively; $\dot{Q}ns_{3c}$, non-shunt flow calculated for 3-compartment model; $(\dot{V}/\dot{Q})_L$, $\dot{V}/\dot{Q}$ ratio for the low $\dot{V}/\dot{Q}$ compartment; $(\dot{V}/\dot{Q})_H$, $\dot{V}/\dot{Q}$ ratio for the high $\dot{V}/\dot{Q}$ compartment; $(\dot{V}/\dot{Q})_{L_j}$ & $(\dot{V}/\dot{Q})_{H_j}$, $j^{th}$ pair of $\dot{V}/\dot{Q}$ ratios for the low and high $\dot{V}/\dot{Q}$ compartments, respectively.

and $O_2$, respectively; $\dot{V}I$ and $\dot{V}E$ are inspired and expired ventilations, respectively; and PB is barometric pressure. In the gas phase, there is the additional constraint:

$$PB = P_{N2} + P_{CO2} + P_{O2} + P_{H2O,} \qquad (4)$$

where $P_{H2O}$ is the saturated water vapour pressure at 37 degrees C.

It is worth noting that, apart from the diffusional equilibration for $CO_2$ and $O_2$ between blood and gas, these equations assume that convection is completely dominant over diffusion so that diffusion within the gas (or blood) phase can be neglected. The model also assumes

that, because $N_2$ is not metabolised and does not reversibly react with blood in large quantities like $CO_2$ and $O_2$, the $N_2$ exchange between blood and gas can be set to zero.

Apart from these relations, the only other one required is a model of the dissociation curves for blood that relates paired values for $P_{CO_2}$ and $P_{O_2}$ to values for blood gas contents ($C_{CO_2}$ and $C_{O_2}$). Various approaches have been adopted by different authors over time, and the current study makes use of a recent numerical model of the oxygen and carbon dioxide dissociation curves for blood [7]. In each case, the patient's reported haemoglobin concentration was used in the model together with an assumed albumin concentration at the lower end of the normal range at 30 gm/L. The $Pa_{CO_2}$, $Pa_{O_2}$ and pH were then used to construct the blood model with the correct acid-base status. Standard values for electrolytes were used as described by [7] except that the plasma chloride concentration was adjusted to provide for electroneutrality. The model described in [7] is represented here by the vector function **g**:

$$\mathbf{c} = \mathbf{g(p)} \tag{5}$$

where $\mathbf{c} = [C_{CO_2}, C_{O_2}]$ and $\mathbf{p} = [P_{CO_2}, P_{O_2}]$.

## Functions

These functions are required to map between values for $\dot{V}/\dot{Q}$, values for the respiratory quotient (R), and paired values for blood gas partial pressures/ contents.

The functions require that the composition of the mixed venous blood is known, that the composition of the inspired gas is known, and that the barometric pressure is known. Busana *et al.* [2] provided the patients' mixed venous oxygen saturations, but no directly measured mixed venous partial pressures or contents. In order to obtain these, we calculated a value for $C\bar{v}_{O_2}$ based on the reported haemoglobin concentration, the reported mixed venous oxygen saturation, and an estimate for the small amount of $O_2$ physically dissolved based on an approximate value for $P\bar{v}_{O_2}$. In the absence of measured mixed venous blood gas contents, Busana *et al.* assumed a respiratory quotient of 0.85 for all patients. The blood gas function, **g**, allowed the estimation of $Ca_{CO_2}$ and $Ca_{O_2}$ from the reported values for $Pa_{CO_2}$ and $Pa_{O_2}$. From this, for each patient we estimated:

$$C\bar{v}_{CO_2} = 0.85(Ca_{O_2} - C\bar{v}_{O_2}) + Ca_{CO_2}. \tag{6}$$

**Function for mapping $\dot{V}/\dot{Q}$ to corresponding respiratory quotient, R.**   This section constructs a function, h, of the form:

$$R = h(\dot{V}/\dot{Q}). \tag{7}$$

The function has a domain of $[0,\infty]$ and a codomain of $[R_{min}, R_{max}]$, where $R_{min}$ is obtained when $\dot{V}/\dot{Q} = 0$ and $R_{max}$ is obtained when $\dot{V}/\dot{Q} \to \infty$. To construct the function, values are required for PB, $PI_{CO_2}$, $PI_{O_2}$, $C\bar{v}_{CO_2}$ and $C\bar{v}_{O_2}$. The function was derived as follows:

Applying mass balance for $N_2$ yields:

$$\dot{V}I(PI_{N_2}/PB) = \dot{V}E(PE_{N_2}/PB), \tag{8}$$

where $PI_{N_2}$ is the inspired $N_2$ partial pressure and $PE_{N_2}$ is the expired $N_2$ partial pressure.

Defining alveolar ventilation as the inspiratory value, $\dot{V} = \dot{V}I$, and rearranging yields:

$$\dot{V}E = (PI_{N_2}/PE_{N_2})\dot{V}. \tag{9}$$

Substitution for $\dot{V}E$ according to *Eq 9* and defining $\dot{V} = \dot{V}I$, allows *Eqs 1 & 2* to be rewritten as:

$$\dot{V}_{CO2} = \dot{Q}(C\bar{v}_{CO2} - Ca_{CO2}) = (PI_{N2}/PE_{N2})\dot{V}P_{CO2}/PB - \dot{V}PI_{CO2}/PB, \text{ and} \tag{10}$$

$$\dot{V}_{O2} = \dot{Q}(C_{O2} - Ca\bar{v}_{O2}) = \dot{V}PI_{O2}/PB - (PI_{N2}/PE_{N2})\dot{V}P_{O2}/PB. \tag{11}$$

Rearranging each yields:

$$(\dot{V}/\dot{Q})((PI_{N2}/PE_{N2})P_{CO2} - PI_{CO2})/PB - (C\bar{v}_{CO2} - \mathbf{g}(Pa_{CO2}, P_{O2})[1, 0]^T) = 0, \text{ and} \tag{12}$$

$$(\dot{V}/\dot{Q})(PI_{O2} - (PI_{N2}/PE_{N2})P_{O2})/PB - (\mathbf{g}(Pa_{CO2}, P_{O2})[0, 1]^T - C\bar{v}_{O2}) = 0, \tag{13}$$

where the blood gas dissociation function, $\mathbf{g}$, has been used to express $Ca_{CO2}$ and $Ca_{O2}$ (*Eq 5*) in terms of $Pa_{CO2}$ and $Pa_{O2}$, and we assume that alveolar $P_{CO2}$ and $P_{O2}$ equal to $Pa_{CO2}$ and $Pa_{O2}$. In order to obtain expressions for $PI_{N2}$ and $PE_{N2}$, the following relationship was used:

$$P_{N2} = PB - P_{CO2} - P_{O2} - P_{H2O,} \tag{14}$$

where $PI_{N2}$ can then be evaluated from $PI_{CO2}$ and $PI_{O2}$, and $PE_{N2}$ expressed in terms of $P_{CO2}$ and $P_{O2}$. These relations now allow *Eq 12* and *Eq 13* to be solved simultaneously for $P_{CO2}$ and $P_{O2}$ by use of a numerical method. Finally, a value for R can be calculated from the relationship:

$$R = (C\bar{v}_{CO2} - \mathbf{g}(P_{CO2}, P_{O2})[1, 0]^T)/(\mathbf{g}(P_{CO2}, P_{O2})[0, 1]^T - C\bar{v}_{O2}), \tag{15}$$

where the blood gas dissociation function, $\mathbf{g}$, was used to calculate $C_{CO2}$ and $C_{O2}$ from $P_{CO2}$ and $P_{O2}$. Alternatively, R may be calculated from the gas phase relationships as:

$$R = ((PI_{N2}/PE_{N2})P_{CO2} - PI_{CO2})/(PI_{O2} - (PI_{N2}/PE_{N2})P_{O2}). \tag{16}$$

**Function for mapping R to $P_{CO2}$ and $P_{O2}$.**   This section describes a vector-valued function, $\mathbf{j}$, of the form:

$$\mathbf{p} = \mathbf{j}(R), \tag{17}$$

where, as previously, $\mathbf{p} = [P_{CO2}, P_{O2}]$. As for the function h, values need to be specified for PB, $PI_{CO2}$, $PI_{O2}$, $C\bar{v}_{CO2}$ and $C\bar{v}_{O2}$. The evaluation of this function then simply involves the simultaneous numerical solution of *Eq 15* and *Eq 16* for $P_{CO2}$ and $P_{O2}$. As for the construction of function h, the expressions for $PI_{N2}$ (in terms of $PI_{CO2}$ and $PI_{O2}$) and $PE_{N2}$ (in terms of $P_{CO2}$ and $P_{O2}$) in *Eq 16* are obtained by use of *Eq 14*.

## Calculation of the candidate $\dot{V}/\dot{Q}$ distributions

Three different types of $\dot{V}/\dot{Q}$ distribution were explored to determine whether specific parameter set(s) could be calculated for them so that they would reproduce the $Pa_{CO2}$ and $Pa_{O2}$ for each patient individually. These are each considered in increasing order of complexity under their individual sections below.

**Estimation of shunt in three-compartment model.**   The three-compartment lung model of Riley and Cournand [5, 6] consists of just one perfused and ventilated compartment that is called the ideal compartment, together with two other compartments consisting of pure shunt (blood flow but not ventilation) and pure deadspace (ventilation but no blood flow),

respectively. The model's special theoretical importance is that for any real $\dot{V}/\dot{Q}$ distribution, no matter how complex, there always exists a corresponding three-compartment model that can exactly replicate the patient's $Pa_{CO2}$ and $Pa_{O2}$. A particular parameter of interest is the non-shunt flow for the three-compartment model, $\dot{Q}ns_{3CM}$. This is the blood flow to the ideal compartment.

First, $\dot{V}_{CO2}$ and $\dot{V}_{O2}$ were calculated directly as:

$$\dot{V}_{CO2} = \dot{Q}t(C\bar{v}_{CO2} - Ca_{CO2}), \text{ and} \tag{18}$$

$$\dot{V}_{O2} = \dot{Q}t(Ca_{O2} - C\bar{v}_{O2}). \tag{19}$$

Next, using function **j**, we calculated the $P_{CO2}$ and $P_{O2}$ of the ideal compartment of the three-compartment model, $Pi_{CO2}$ and $Pi_{O2}$, as:

$$[Pi_{CO2}, Pi_{O2}] = \mathbf{j}(0.85). \tag{20}$$

We then calculated the associated blood-gas contents through function **g**:

$$[Ci_{CO2}, Ci_{O2}] = \mathbf{g}([Pi_{CO2}, Pi_{O2}]), \tag{21}$$

where $Ci_{CO2}$ and $Ci_{O2}$ are blood contents of ideal compartment.

From these values, $\dot{Q}ns_{3CM}$ was estimated by either of the following two relations:

$$\dot{Q}ns_{3CM} = \dot{V}_{CO2}/(C\bar{v}_{CO2} - Ci_{CO2}), \text{ or} \tag{22}$$

$$\dot{Q}ns_{3CM} = \dot{V}_{O2}/(Ci_{O2} - C\bar{v}_{O2}). \tag{23}$$

Importantly, $\dot{Q}ns_{3CM}$ forms a minimum value for the non-shunt flow, as it is all used optimally to perfuse the ideal compartment of the model. From this, the shunt flow for the three-compartment model, $\dot{Q}s_{3CM}$, can be calculated by subtracting the shunt flow from the total cardiac output, $\dot{Q}t$.

The shunt fraction assumption proposed by Busana *et al.*, $\dot{Q}s_{BM}$ (subscript 'BM' refers to the model of Busana *et al.*), is *not* that associated with the three-compartment model, but instead is proportional to the non-aerated lung fraction as quantified by computer tomography (CT). Subtraction of $\dot{Q}s_{BM}$ from $\dot{Q}t$ yields the non-shunt flow that perfuses the remaining aerated lung, $\dot{Q}ns_{BM}$. The distinction between $\dot{Q}ns_{BM}$ and $\dot{Q}ns_{3CM}$ allowed an overall efficiency for the use of the perfusion, $E\dot{Q}ns_{BM}$, for the ventilated and perfused compartments to be defined as $\dot{Q}ns_{3CM}/\dot{Q}ns_{BM}$. As $\dot{Q}ns_{3CM}$ is the minimum possible non-shunt flow, this ratio may be seen as the ratio of the minimum-to-actual blood flow required to deliver the gas exchange, and therefore should be less than, or equal to, one. (It is also equivalent to the ratio of the actual-to-maximum $O_2$ consumption or $CO_2$ production that is possible with a blood flow of $\dot{Q}ns_{BM}$.) From this, it follows immediately that, if $\dot{Q}ns_{3CM} > \dot{Q}ns_{BM}$, then $\dot{Q}ns_{BM}$ is too low to support the gas-exchange required to produce the patient's measured $Pa_{O2}$ and $Pa_{CO2}$. This provides a basis for our test of whether $\dot{V}/\dot{Q}$ distributions exist for the shunt flows proposed by Busana *et al.*

**Estimation of $\dot{V}/\dot{Q}$ values for model with two perfused compartments.** This section explored whether a pair of $\dot{V}/\dot{Q}$ compartments, one with low $\dot{V}/\dot{Q}$, $(\dot{V}/\dot{Q})_L$, and one with high $\dot{V}/\dot{Q}$, $(\dot{V}/\dot{Q})_H$, could reproduce a patient's values for $Pa_{CO2}$ and $Pa_{O2}$, assuming that the combined total perfusion of the two compartments was equal to $\dot{Q}ns_{BM}$, (the shunt fraction

assumption) which was calculated from the non-aerated lung fraction, $F_{NAL}$, as follows:

$$\dot{Q}ns_{BM} = (1 - F_{NAL})\dot{Q}t, \tag{24}$$

where the values for $F_{NAL}$ are those provided by Busana *et al.* [2]. Expressions for total $\dot{V}_{CO2}$ and $\dot{V}_{O2}$ may be written by summing the $CO_2$ production from the low and high $\dot{V}/\dot{Q}$ compartments, and the $O_2$ consumption from the low and high $\dot{V}/\dot{Q}$ compartments, respectively, as follows:

$$\dot{V}_{CO2} = (\dot{Q}ns_{BM}f)(C\bar{v}_{CO2} - \mathbf{g}(\mathbf{j}(h((\dot{V}/\dot{Q})_L))))[1,0]^T) + (\dot{Q}ns_{BM}(1-f))(C\bar{v}_{CO2} - \mathbf{g}(\mathbf{j}(h((\dot{V}/\dot{Q})_H))))[1,0]^T), \text{ and} \tag{25}$$

$$\dot{V}_{O2} = (\dot{Q}ns_{BM}f)(\mathbf{g}(\mathbf{j}(h((\dot{V}/\dot{Q})_L))))[0,1]^T - C\bar{v}_{O2}) + (\dot{Q}ns_{BM}(1-f))(\mathbf{g}(\mathbf{j}(h((\dot{V}/\dot{Q})_H))))[0,1]^T - C\bar{v}_{O2}), \tag{26}$$

where f is the fraction of blood flow to the low $\dot{V}/\dot{Q}$ compartment, and (1 –f) is the fraction of blood flow to the high $\dot{V}/\dot{Q}$ compartment.

*Eqs 25* and *26* have three unknown variables: $(\dot{V}/\dot{Q})_L$, $(\dot{V}/\dot{Q})_H$ and f. Thus, the general approach taken was to choose a value for $(\dot{V}/\dot{Q})_L$ and then solve the equations simultaneously to obtain values for $(\dot{V}/\dot{Q})_H$ and f that are associated with the particular value for $(\dot{V}/\dot{Q})_L$. However, for any particular choice for $(\dot{V}/\dot{Q})_L$, in general there is no guarantee that a solution will exist.

In order to establish possible bounds for $(\dot{V}/\dot{Q})_L$ within which solutions for *Eqs 25* and *26* may exist, it is first worth noting that, if $(\dot{V}/\dot{Q})_L = 0$, then all gas exchange has to arise from the $(\dot{V}/\dot{Q})_H$ compartment. In this scenario $(\dot{V}/\dot{Q})_H$ must equal the value for the ideal compartment from the associated three-compartment model (and $\dot{Q}ns_{BM}(1—f) = \dot{Q}ns_{3CM}$). As $(\dot{V}/\dot{Q})_L$ is increased above zero, it will start to contribute to gas exchange with a value for R that is below the overall R value for the patient. Consequently gas exchange from the $(\dot{V}/\dot{Q})_H$ compartment will require a value for R above that for the overall patient. Therefore the value for $(\dot{V}/\dot{Q})_H$ must increase above the that of the ideal compartment from the associated three-compartment model. This reasoning suggests that a possible maximum for $(\dot{V}/\dot{Q})_L$ may arise as the value for $(\dot{V}/\dot{Q})_H \rightarrow \infty$. Consequently, we first solved *Eqs 25* and *26* for $(\dot{V}/\dot{Q})_L$ and f under the condition $(\dot{V}/\dot{Q})_H \rightarrow \infty$, and subsequently sought solutions for *Eqs 25* and *26* (in terms of $(\dot{V}/\dot{Q})_H$ and f) for values of $(\dot{V}/\dot{Q})_L$ between 0 and this putative maximum.

**Estimation of $\dot{V}/\dot{Q}$ distributions for model with multiple perfused compartments.** The final step is to attempt to construct models with multiple perfused units that could replicate the gas exchange of the patient. The preceding section suggests one possible way forward is to construct them from pairs of $\dot{V}/\dot{Q}$ compartments, where each pairing has one value below and one value above the ideal $\dot{V}/\dot{Q}$, $(\dot{V}/\dot{Q})i$. Symmetry suggests a natural pairing between the j$^{th}$ low, $(\dot{V}/\dot{Q})L_j$, and the j$^{th}$ high, $(\dot{V}/\dot{Q})H_j$, compartments of the form:

$$(\dot{V}/\dot{Q})L_j/(\dot{V}/\dot{Q})i = (\dot{V}/\dot{Q})i/(\dot{V}/\dot{Q})H_j, \tag{27}$$

which, when rearranged gives:

$$(\dot{V}/\dot{Q})H_j = ((\dot{V}/\dot{Q})i)^2/(\dot{V}/\dot{Q})L_j. \tag{28}$$

Following the approach of Busana *et al.* [2], we constructed a set of N evenly-spaced sub-intervals on a logarithmic basis within an interval $[(\dot{V}/\dot{Q})i/k, (\dot{V}/\dot{Q})i]$ for the low $\dot{V}/\dot{Q}$ units, and a further N evenly-spaced sub-intervals on a logarithmic basis within an interval $[(\dot{V}/\dot{Q})i, k(\dot{V}/\dot{Q})i]$ for the high $\dot{V}/\dot{Q}$ units. We chose a value of 50 for N and 100 for k. Values for $(\dot{V}/\dot{Q})L_j$ and $((\dot{V}/\dot{Q})H_j$ were then allocated as the (logarithmically) central values for each of the sub-intervals below and above $(\dot{V}/\dot{Q})i$, respectively.

For each $\dot{V}/\dot{Q}$ unit, function h (*Eq 7*) provides the associated value for R; for each value of R, function **j** (*Eq 17*) provides the unit's associated $P_{CO2}$ and $P_{O2}$; and for each pair of $P_{CO2}$ and $P_{O2}$ values, function **g** (*Eq 5*) provides the unit's associated blood gas contents, $C_{CO2}$ and $C_{O2}$. For each pair of units, we can solve for the fractional blood flow to the lower unit, $f_j$, in relation to the combined blood flow to both units that generates the overall R value for the patient:

$$R = (f_j(C\bar{v}_{CO2} - CL_{CO2,j}) + (1 - f_j)(C\bar{v}_{CO2} - CH_{CO2,j}))/(f_j(CL_{O2,j} - C\bar{v}_{O2}) + (1 - f_j)(CH_{O2,j} - C\bar{v}_{O2})),$$

(29)

where $CH_{CO2, j}$ and $CL_{CO2, j}$ are the $j^{th}$ pair of blood $CO_2$ contents for the *j*th pair of high and low $\dot{V}/\dot{Q}$ units, respectively; and $CH_{O2, j}$ and $CL_{O2,j}$ are the $j^{th}$ pair of blood $O_2$ contents for the *j*th pair of high and low $\dot{V}/\dot{Q}$ units, respectively.

By considering the $O_2$ consumption per unit of blood flow for the pair of units relative to that which would occur per unit of blood flow to the ideal compartment of the three-compartment model, an efficiency for the use of the perfusion to the $j^{th}$ pair of units ($E\dot{Q}_j$) may be calculated as:

$$E\dot{Q}_j = (f_j(CL_{O2,j} - C\bar{v}_{O2}) + (1 - f_j)(CH_{O2,j} - C\bar{v}_{O2}))/(Ci_{O2} - C\bar{v}_{O2}).$$

(30)

From this construction, any distribution that obeys the relationship:

$$E\dot{Q}ns_{BM} = \sum^j(\dot{Q}_j E\dot{Q}_j)/\dot{Q}ns_{BM},$$

(31)

where $E\dot{Q}ns_{BM}$ is the overall circulatory efficiency for the patient given by $\dot{Q}ns_{3CM}/\dot{Q}ns_{BM}$, should reproduce the patient's arterial blood gas values precisely.

One approach to finding solutions to *Eq 31* is to apply a parameterised probability distribution over the interval containing the values for $(\dot{V}/\dot{Q})L_j$ to distribute $\dot{Q}ns_{BM}$ over the pairs of values for $\dot{V}/\dot{Q}$. We chose a bounded probability function, the beta distribution, and partitioned it into N equal divisions on the interval [0,1]. The beta distribution has two shape parameters, α and β, where α>0 and β>0. For our purposes, it was convenient to reparameterize this in the form:

$$\alpha = \theta, \text{ and}$$

(32)

$$\beta = \mu - \theta,$$

(33)

where μ is a constant greater than zero, a value of 10 was initially chosen, and $0 < \theta < \mu$. This parameterisation ensured that the peak for the probability distribution gradually migrated rightwards on the interval [0,1] as θ increased. The values for $\dot{Q}_j$ could then be written using

the cumulative density function for the beta distribution, $\text{beta}_{\text{cdf}}(\alpha, \beta, x)$, as follows:

$$\dot{Q}_j = (\text{beta}_{\text{cdf}}(\theta, (\mu - \theta), j/\text{N}) - \text{beta}_{\text{cdf}}(\theta, (\mu - \theta), (j-1)/\text{N}))\dot{Q}\text{ns}_{\text{BM}}. \quad (34)$$

These may be substituted into *Eq 31* to yield:

$$\text{E}\dot{Q}\text{ns}_{\text{BM}} = \sum^{j}(\text{beta}_{\text{cdf}}(\theta, (\mu - \theta), j/\text{N}) - \text{beta}_{\text{cdf}}(\theta, (\mu - \theta), (j-1)/\text{N}))\text{E}\dot{Q}_j. \quad (35)$$

Setting $\text{E}\dot{Q}\text{ns}_{\text{BM}}$ equal to the overall efficiency of gas exchange for the patient, the equation could be solved for $\theta$ to provide a possible multi-compartment $\dot{V}/\dot{Q}$ distribution for the patient. Other distributions could be calculated by varying the value of $\mu$ for the beta distribution, or potentially could be obtained by employing other distributions in place of the beta distribution.

## Results

### Shunt flow calculation for each patient using the three-compartment model

The essential patient data from Busana *et al.* [2] are given in Table 1 together with the results obtained for each patient from fitting a three-compartment model to their gas exchange data. Patients 2, 3 and 5 were relatively similar in terms of their blood gases. Patient 1 had a substantially higher $\text{Pa}_{\text{O2}}$ than the other patients, and patient 4 had a substantially lower $\text{Pa}_{\text{CO2}}$ than the other patients. The $\text{P}_{\text{CO2}}$ and $\text{P}_{\text{O2}}$ values for the ideal compartment for each patient reflected these initial differences.

In the case of patient 1, the required perfusion to the ideal compartment of the three-compartment model, $\dot{Q}\text{ns}_{\text{3CM}}$, exceeded the hypothesised value, $\dot{Q}\text{ns}_{\text{BM}}$, as obtained from the total cardiac output and fraction of non-ventilated lung ($\text{F}_{\text{NAL}}$ estimated from CT in Busana *et al*). Given that $\dot{Q}\text{ns}_{\text{3CM}}$ is the minimum possible value for perfusion for the required gas exchange, this finding means that, for this particular patient, the assumption that the shunt fraction is equivalent to fraction of non-aerated lung quantified by CT cannot be valid. The true shunt fraction for this particular patient has to be below that calculated from the fraction of non-ventilated lung. For all other patients $\dot{Q}\text{ns}_{\text{BM}}$ substantially exceeded $\dot{Q}\text{ns}_{\text{3CM}}$, allowing the exploration of possible $\dot{V}/\dot{Q}$ distributions to proceed. The ratio of $\dot{Q}\text{ns}_{\text{3CM}}:\dot{Q}\text{ns}_{\text{BM}}$ is shown as an efficiency, $\text{E}\dot{Q}\text{ns}_{\text{BM}}$, for the use of the blood flow that is being used to support gas exchange, and was less than 1 for patients 2–5.

### Estimation of $\dot{V}/\dot{Q}$ pairs (two perfused compartment model) when total perfusion is equal to $\dot{Q}\text{ns}_{\text{BM}}$

Figs 2 and 3 illustrate the results for estimating pairs of low and high values for $\dot{V}/\dot{Q}$ that will replicate the $\text{Pa}_{\text{CO2}}$ and $\text{Pa}_{\text{O2}}$ values for each patient. For all pairs, the numerical error between the calculated and actual values for $\text{Pa}_{\text{CO2}}$ and $\text{Pa}_{\text{O2}}$ in the patients was very low, and always $< 1\text{x}10^{-2}$ mmHg. This is a very major improvement in precision (2–3 orders of magnitude) compared with the multiple random simulations approach that was employed by Busana *et al*, and it demonstrates that, assuming $\dot{Q}\text{ns}_{\text{BM}}$ is the non-shunt blood flow for the aerated lung, there are multiple (infinite) solutions for $\dot{V}/\dot{Q}$ distributions that match the patients' $\text{Pa}_{\text{CO2}}$ and $\text{Pa}_{\text{O2}}$ precisely. Fig 2 illustrates that as the $\dot{V}/\dot{Q}$ for the low $\dot{V}/\dot{Q}$ compartment increased, there was a progressive increase in the low $\dot{V}/\dot{Q}$ compartment's share of total blood flow (left-axis) and a progressive rise in $\dot{V}/\dot{Q}$ for the high $\dot{V}/\dot{Q}$ compartment (right-axis). As a

**Table 1. Patient data and results from fitting three-compartment model of gas exchange.**

| Patient | 1 | 2 | 3 | 4 | 5 |
|---|---|---|---|---|---|
| Hb /g·L$^{-1}$ | 95 | 100 | 117 | 86 | 114 |
| **Inspired gas** | | | | | |
| PI$_{CO2}$ /mmHg(kPa) | 0.0(0.0) | 0.0(0.0) | 0.0(0.0) | 0.0(0.0) | 0.0(0.0) |
| PI$_{O2}$ /mmHg(kPa) | 641.7(85.5) | 606.1(80.8) | 570.4(76.0) | 570.4(76.0) | 427.8(57.0) |
| **Arterial blood** | | | | | |
| Pa$_{CO2}$ /mmHg(kPa) | 71.0(9.5) | 79.0(10.5) | 69.5(9.3) | 42.0(5.6) | 83.3(11.1) |
| Pa$_{O2}$ /mmHg(kPa) | 105.0(14.0) | 62.0(8.3) | 65.3(8.7) | 64.0(8.5) | 62.5(8.3) |
| Ca$_{CO2}$ /L$_{STPD}$·L$^{-1}$ | 0.8685 | 0.8148 | 0.7759 | 0.5673 | 0.8097 |
| Ca$_{O2}$ /L$_{STPD}$·L$^{-1}$ | 0.1299 | 0.1231 | 0.1479 | 0.1107 | 0.1395 |
| pHa | 7.390 | 7.310 | 7.360 | 7.430 | 7.290 |
| **Mixed Venous blood** | | | | | |
| P$\bar{v}_{CO2}$/mmHg(kPa) | 79.1(10.5) | 84.8(11.3) | 74.7(10.0) | 46.7(6.2) | 89.3(11.9) |
| P$\bar{v}_{O2}$/mmHg(kPa) | 38.4(5.1) | 41.0(5.5) | 42.5(5.7) | 35.3(4.7) | 42.6(5.7) |
| C$\bar{v}_{CO2}$/L$_{STPD}$·L$^{-1}$ | 0.9019 | 0.8383 | 0.7999 | 0.5926 | 0.8346 |
| C$\bar{v}_{O2}$/L$_{STPD}$·L$^{-1}$ | 0.0906 | 0.0954 | 0.1196 | 0.0809 | 0.1102 |
| **Ideal compartment in a three-compartment model** | | | | | |
| Pi$_{CO2}$ /mmHg(kPa) | 67.6(9.0) | 71.8(9.6) | 63.2(8.4) | 37.3(5.0) | 76.1(10.1) |
| Pi$_{O2}$ /mmHg(kPa) | 572.1(76.3) | 531.8(70.9) | 504.9(67.3) | 531.8(70.9) | 348.0(46.4) |
| Ci$_{CO2}$ /L$_{STPD}$·L$^{-1}$ | 0.8587 | 0.7917 | 0.7530 | 0.5479 | 0.7852 |
| Ci$_{O2}$ /L$_{STPD}$·L$^{-1}$ | 0.1470 | 0.1525 | 0.1749 | 0.1335 | 0.1659 |
| **Perfusion** | | | | | |
| $\dot{Q}$t/L·min$^{-1}$ | 8.060 | 9.490 | 10.400 | 9.450 | 10.550 |
| $\dot{Q}$s$_{BM}$/L·min$^{-1}$ | 3.224 | 3.701 | 3.744 | 2.552 | 2.743 |
| $\dot{Q}$ns$_{BM}$/L·min$^{-1}$ | 4.836 | 5.789 | 6.656 | 6.899 | 7.807 |
| $\dot{Q}$s$_{3CM}$/L·min$^{-1}$ | 2.447 | 4.897 | 5.079 | 4.086 | 5.001 |
| $\dot{Q}$ns$_{3CM}$/L·min$^{-1}$ | 5.613 | 4.593 | 5.321 | 5.364 | 5.549 |
| EQ$\dot{n}$s$_{BM}$ | (1.161) | 0.793 | 0.799 | 0.778 | 0.711 |

PI$_{CO2}$ & PI$_{O2}$, inspired P$_{CO2}$ & P$_{O2}$, respectively; Hb, haemglobin concentration; Pa$_{CO2}$ & Pa$_{O2}$, arterial P$_{CO2}$ and P$_{O2}$, respectively; Ca$_{CO2}$ & Ca$_{O2}$, arterial gas contents for $CO_2$ and $O_2$, respectively; pHa, arterial pH; P$\bar{v}_{CO2}$ & P$\bar{v}_{O2}$, mixed venous P$_{CO2}$ and P$_{O2}$, respectively; C$\bar{v}_{CO2}$ & C$\bar{v}_{O2}$, mixed venous gas contents for $O_2$ and $O_2$, respectively; Pi$_{CO2}$ & Pi$_{O2}$, P$_{CO2}$ and P$_{O2}$ for ideal compartment of three-compartment model, respectively; Ci$_{CO2}$ & Ci$_{O2}$, blood gas contents for $CO_2$ and $O_2$ for ideal compartment of three-compartment model, respectively; $\dot{Q}$t, total cardiac output; $\dot{Q}$s$_{BM}$ & $\dot{Q}$ns$_{BM}$ shunt flow to non-aerated lung and corresponding non-shunt flow from Busana *et al.*, respectively; $\dot{Q}$s$_{3CM}$ & $\dot{Q}$ns$_{3CM}$ shunt flow for three-compartment model of lung and corresponding non-shunt flow, respectively; EQ$\dot{n}$s$_{BM}$, perfusion efficiency relative to maximal value associated with three-compartment model.

higher $\dot{V}/\dot{Q}$ value for the low $\dot{V}/\dot{Q}$ compartment was associated with a higher $\dot{V}/\dot{Q}$ value for the high $\dot{V}/\dot{Q}$ compartment, we calculated putative maximal values of $\dot{V}/\dot{Q}$ for the low $\dot{V}/\dot{Q}$ compartment by assuming that, in the high $\dot{V}/\dot{Q}$ compartment, the blood was maximally oxygenated and had all the $CO_2$ removed (i.e. $(\dot{V}/\dot{Q})_H \rightarrow \infty$). The maximal values for $\dot{V}/\dot{Q}$ obtained for the low $\dot{V}/\dot{Q}$ compartment in this manner were 0.060, 0.064, 0.084 and 0.078, for patients 2, 3, 4 and 5, respectively.

Fig 3 illustrates the increase in the total combined ventilation to both compartments as the value for $\dot{V}/\dot{Q}$ associated with the low $\dot{V}/\dot{Q}$ compartment increased. In contrast to perfusion, where there was a fixed value for $\dot{Q}$ns$_{BM}$ for each patient, Busana *et al* did not provide

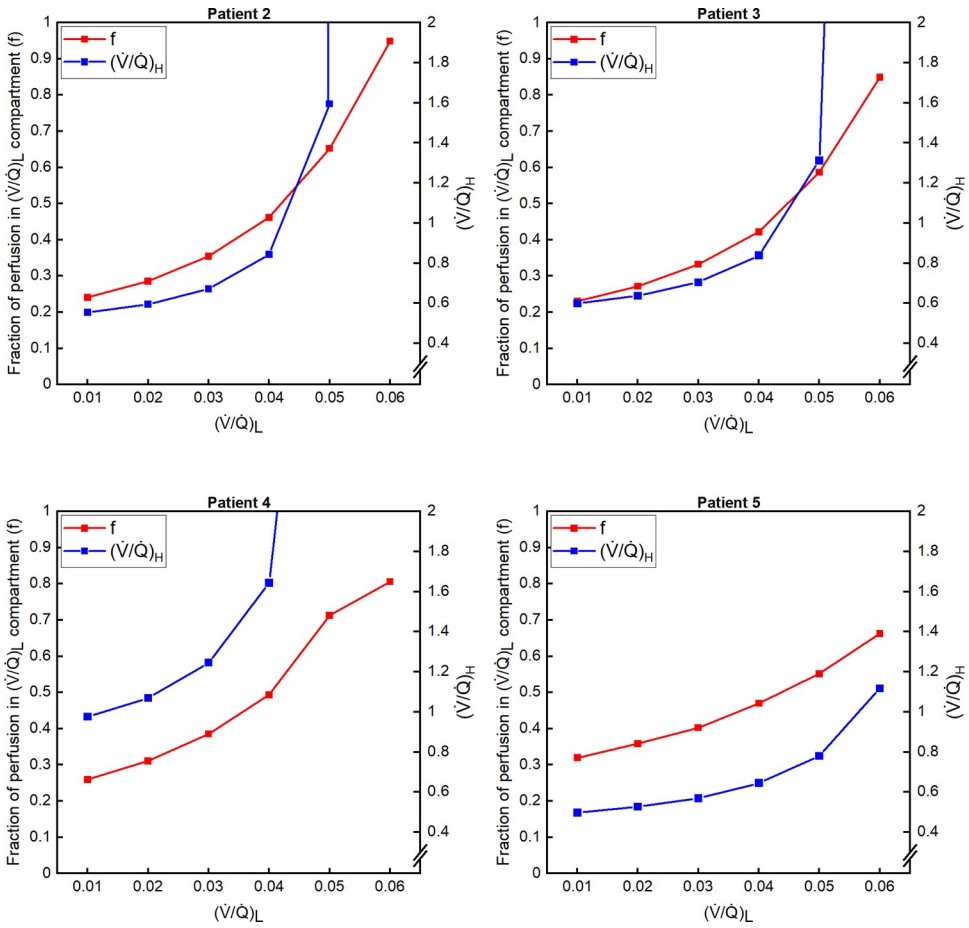

**Fig 2. Fraction of total perfusion to the low ventilation-perfusion ($\dot{V}/\dot{Q}$) compartment and $\dot{V}/\dot{Q}$ value for high $\dot{V}/\dot{Q}$ compartment in relation to $\dot{V}/\dot{Q}$ value for low $\dot{V}/\dot{Q}$ compartment.** Results shown for patients 2–5. Some values off-scale for higher values of $\dot{V}/\dot{Q}$ for the low $\dot{V}/\dot{Q}$ compartment. Differences between measured and calculated arterial partial pressures for $CO_2$ and $O_2 < 1 \times 10^{-2}$ mmHg for all distributions illustrated. f, fraction of total perfusion to the low $\dot{V}/\dot{Q}$ compartment; $(\dot{V}/\dot{Q})_L$, $\dot{V}/\dot{Q}$ ratio for the low $\dot{V}/\dot{Q}$ compartment; $(\dot{V}/\dot{Q})_H$, $\dot{V}/\dot{Q}$ ratio for the high $\dot{V}/\dot{Q}$ compartment.

information to allow the specification of a fixed value for alveolar ventilation for each patient. Were such information to be available, then the monotonic increasing nature of the ventilation in Fig 3 suggests that there would be only one pair of $\dot{V}/\dot{Q}$ units that could satisfy constraints on the values for both $\dot{Q}ns_{BM}$ and alveolar ventilation.

## Ventilation-perfusion distributions assuming multiple perfused compartment pairs

Fig 4 illustrates a $\dot{V}/\dot{Q}$ distribution for each patient based on the beta distribution with an assumed shape parameter μ (= α + β) of 10. For each patient, the residual numerical error in the estimate for their $Pa_{CO2}$ and $Pa_{O2}$ was $10^{-3}$ mmHg or less. Fig 5 (upper and bottom left-hand panels) illustrates the effect of varying the shape parameter (μ = 5, 10, 20) on the distribution for patient 4. Higher values of μ were associated with steeper peaks and higher total overall values for ventilation. Finally, the right-hand lower panel of Fig 5 illustrates the shape of the

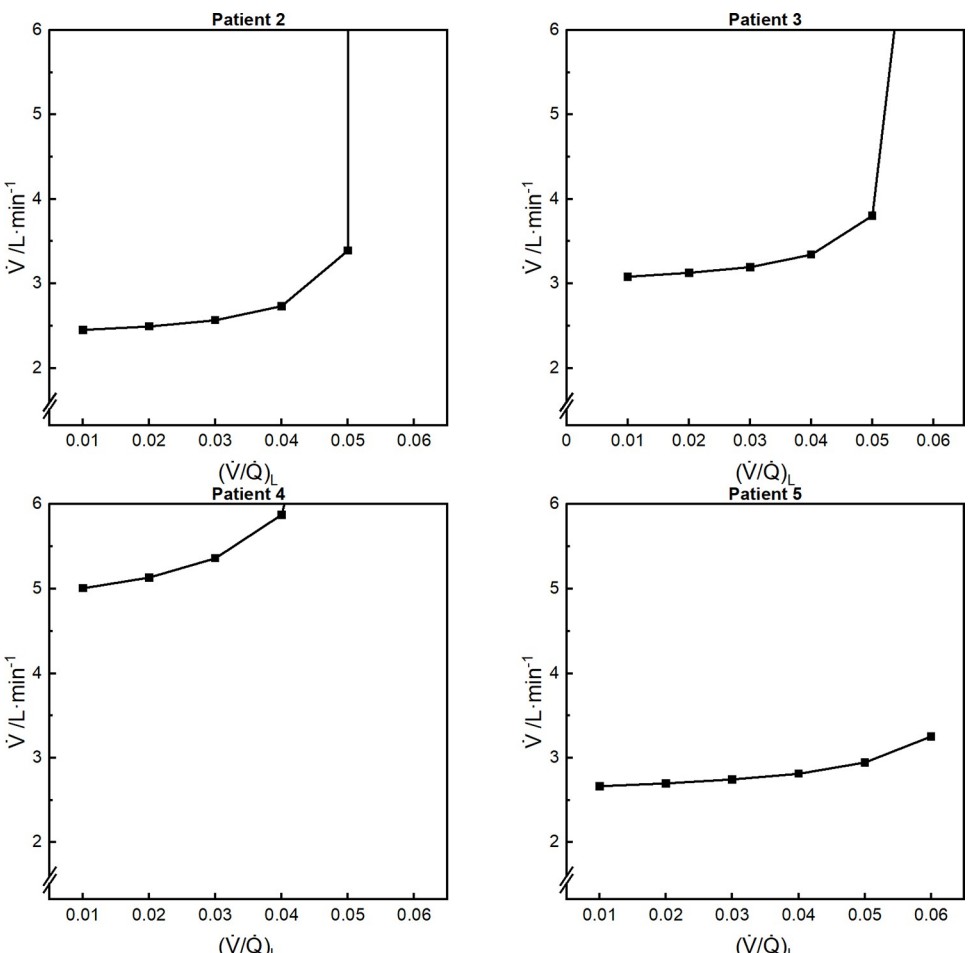

**Fig 3. Total ventilation as a function of $(\dot{V}/\dot{Q})_L$.** Results shown for patients 2–5. Some values off-scale for higher values of $(\dot{V}/\dot{Q})_L$. Differences between measured and calculated arterial partial pressures for $CO_2$ and $O_2 < 1x10^{-2}$ mmHg for all distributions illustrated. $\dot{V}$, total ventilation.

distribution for a hypothetical healthy person breathing air with $P\bar{v}_{CO2} = 46$ mmHg, $C\bar{v}_{O2} = 40$ mmHg, $Pa_{CO2} = 40$ mmHg, $Pa_{O2} = 96$ mmHg, and a presumed efficiency of use for the blood of 0.987. Here the separate $\dot{V}/\dot{Q}$ peaks in the high and low $\dot{V}/\dot{Q}$ regions have been lost and the calculation has resulted in a single central peak for the $\dot{V}/\dot{Q}$ distribution.

## Discussion

Busana *et al.* [2] reported on five patients with severe COVID-19 pneumonia; they hypothe-sised that the shunt blood flow within the lungs was proportional to the fraction of non-aerated lung tissue quantified by CT, and they subsequently employed a random simulation approach that identified candidate $\dot{V}/\dot{Q}$ distributions in four out of five patients that replicated the patients' $Pa_{CO2}$ and $Pa_{O2}$ values with limited precision. The purpose of the present study was to seek higher precision solutions that would then be within the accuracy of the blood-gas data, by developing a method that enabled direct calculation of candidate $\dot{V}/\dot{Q}$ distributions without using their random simulation approach. For the patient for whom Busana et al could find no $\dot{V}/\dot{Q}$ distributions, we were able to show that the shunt flow assumption resulted in a

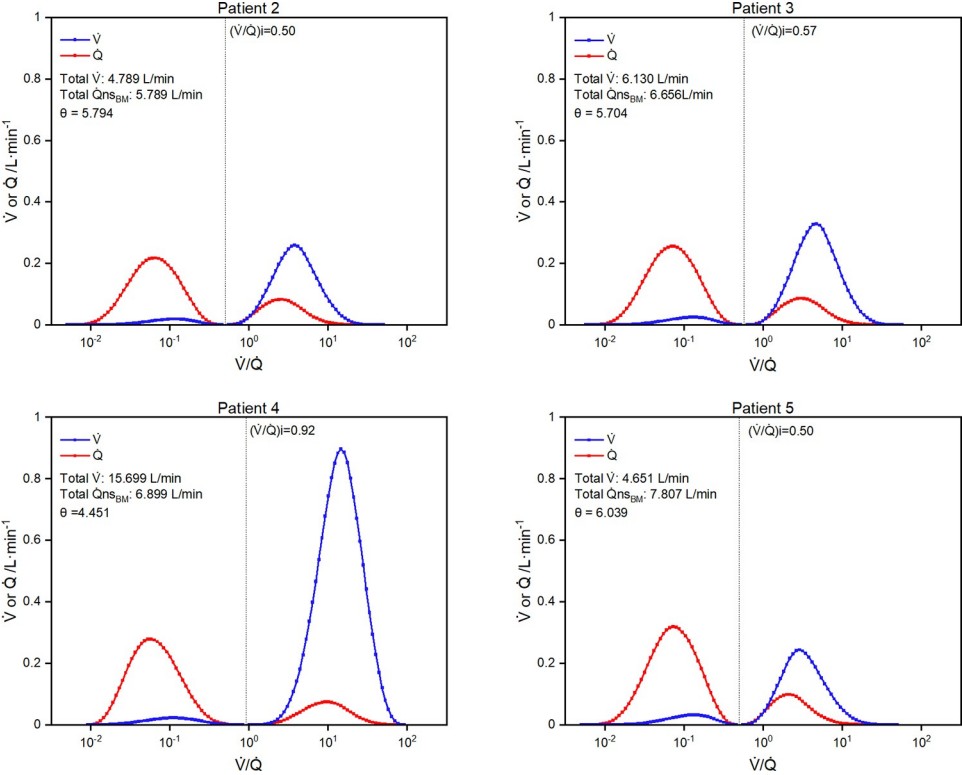

**Fig 4. Multi-compartment distributions for $\dot{V}/\dot{Q}$.** Results shown for patients 2–5. Distribution based on beta distribution with parameter $\mu = 10$, as defined in *Eq 33*. Differences between measured and calculated arterial partial pressures for $CO_2$ and $O_2 < 1 \times 10^{-3}$ mmHg for all distributions. $\dot{Q}$, perfusion; $\dot{Q}ns_{BM}$, total non-shunt perfusion where shunt fraction has been based on the non-aerated lung fraction; $\theta$, parameter (estimated) for beta distribution as defined in *Eq 32*; $(\dot{V}/\dot{Q})$i, $\dot{V}/\dot{Q}$ value for ideal compartment for three-compartment model of lung.

remaining non-shunt blood flow that was simply too low to support the gas exchange required, and therefore no solutions exist. For the other four patients, we have demonstrated that multiple, indeed infinite sets, of potential $\dot{V}/\dot{Q}$ distributions exist that were capable of reproducing the patients' $Pa_{CO2}$ and $Pa_{O2}$ values precisely. We have identified a few of these solutions, both in the form of single pairs of $\dot{V}/\dot{Q}$ compartments and in the form of distributions of multiple simultaneous pairs of $\dot{V}/\dot{Q}$ compartments.

Why the approach of Busana *et al.* did not lead to solutions very close to the patients' $Pa_{CO2}$ and $Pa_{O2}$ values is not entirely clear. One possibility is that $10^6$ randomly chosen simulations is simply an insufficient number. This may appear strange, but Busana *et al.* report that their distribution has 5 parameters, and if each random choice of one parameter were to be permuted with every random choice of the other, then that would only allow 16 random choices for each parameter ($16^5 \approx 10^6$). While this is possible, we suspect a more likely reason is that it is difficult to design a strategy for ensuring that any random process of generating distributions provides an even remotely even coverage of possible values for $Pa_{CO2}$ and $Pa_{O2}$. Evidence of this is very clear from the figure numbered six in their paper, which is an arterial $P_{CO2}/P_{O2}$ diagram, and shows regions where the density of possible $\dot{V}/\dot{Q}$ distributions is very high, and other regions that are covered by no possible $\dot{V}/\dot{Q}$ distributions at all. The present study obviates this difficulty by showing that it is possible directly to construct putative $\dot{V}/\dot{Q}$ distributions

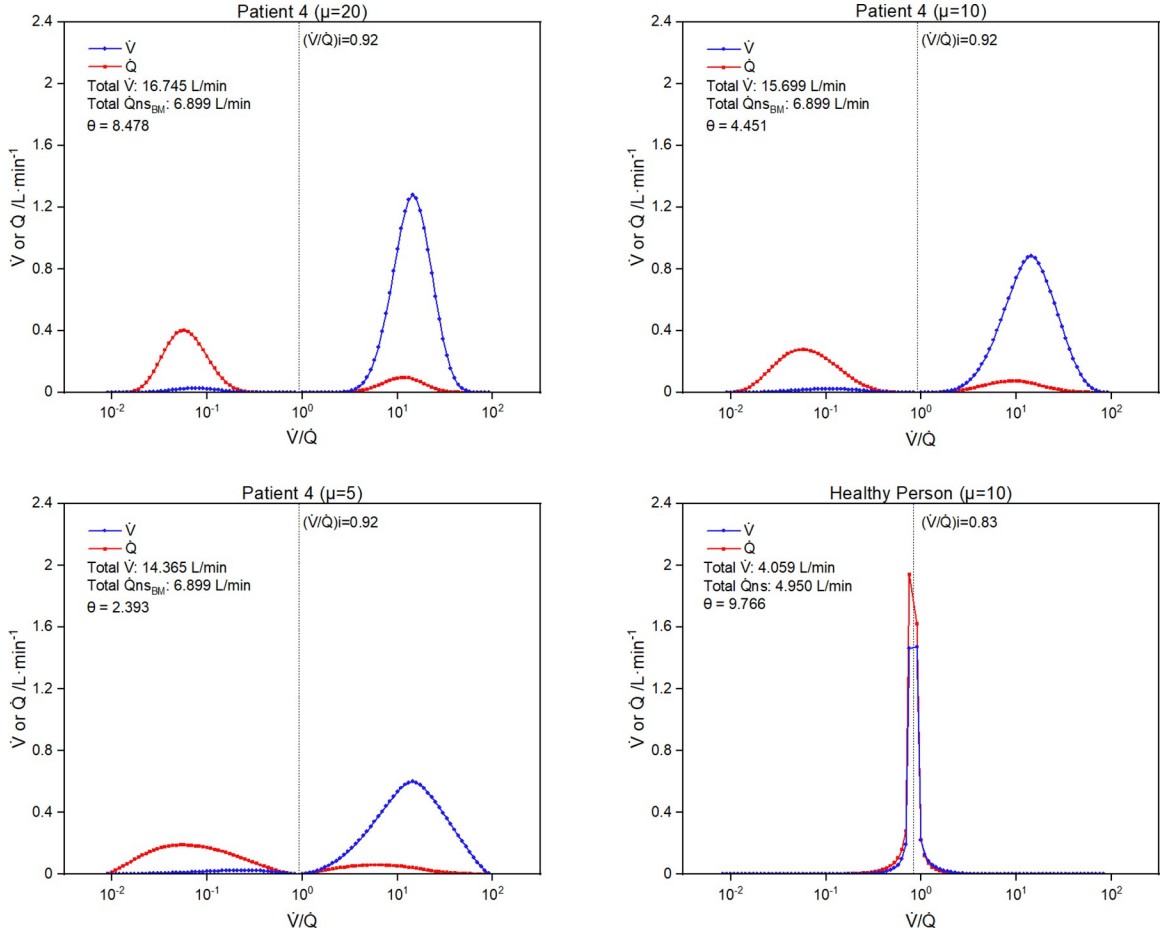

**Fig 5. Variations in multi-compartment $\dot{V}/\dot{Q}$ distribution generated by changing parameters of the beta distribution and by changing physiological status.** Results for top-left, top-right, and bottom-left panels are for patient 4 with $\mu$ = 20, 10 and 5, respectively. Results for bottom right panel are for standard blood gas values for a healthy individual with $\mu$ = 10. Differences between measured and calculated arterial partial pressures for $CO_2$ and $O_2 < 1\times10^{-3}$ mmHg for all distributions. $\dot{Q}$ns, total non-shunt blood flow.

that are guaranteed to reproduce the $Pa_{CO_2}$ and $Pa_{O_2}$ of individual patients based on the classical mass balance relations that have been established for gas exchange in the lung.

Busana *et al.* did not provide data concerning the patient's ventilation. The simulations constructed here vary substantially in relation to total alveolar ventilation (see Fig 3). If this ventilation were known for a patient, then it should be possible to use that information to select the particular pair of high and low $\dot{V}/\dot{Q}$ compartments consistent with the patient's gas exchange, or alternatively select the particular shape parameter for the beta distribution ($\mu$) based on the ventilation (see Fig 5).

Experimentally, the physiological approach that comes closest to providing a $\dot{V}/\dot{Q}$ distribution for a patient is the multiple inert gas elimination technique (MIGET) developed by Wagner and West [8]. This involves infusing into a vein a range of dissolved gases and measuring their retention and excretion ratios. In relation to the present study, it is of note that a retention for a very insoluble gas in the arterial blood would effectively provide the fraction of total blood flow that is pure shunt, and an excretion for a very soluble gas would effectively provide the fraction of total ventilation that is alveolar. Therefore, using the methodology developed in the present study, such information in combination with the $P_{CO_2}$ and $P_{O_2}$ values would be

sufficient to calculate directly a single pair of low and high $\dot{V}/\dot{Q}$ compartments, or alternatively calculate a single beta distribution for $\dot{V}/\dot{Q}$, that is most representative of the $\dot{V}/\dot{Q}$ distribution within a patient.

Turning from the methodology to the results, what is evident from our solutions for the $\dot{V}/\dot{Q}$ distributions from both the methods we employed (Figs 2 and 4) is that the assumption that the shunt fraction is proportional to the fraction of non-ventilated lung (proposed by Busana *et al*) led to compartments with extremely low estimates for $\dot{V}/\dot{Q}$. For example, the maximum possible values for $\dot{V}/\dot{Q}$ for the $(\dot{V}/\dot{Q})_L$ compartment were below 0.09 for all four patients. Similarly for the beta distribution, a considerable fraction of the total non-shunt perfusion was assigned to units of very low $\dot{V}/\dot{Q}$. In relation to this, we have considerable reservations as to whether the classical theory developed in relation to $\dot{V}/\dot{Q}$ distribution is actually applicable when very low $\dot{V}/\dot{Q}$ ratios are calculated for patients who are breathing gas containing a high inspired fraction of $O_2$, such as in these COVID-19 patients. This is because the classical theory makes no allowance for the movement of gases by diffusion other than that across the alveolar membrane. An extreme example would be apnoeic oxygenation, where the lung is motionless and the airway connected to 100% $O_2$. Classical $\dot{V}/\dot{Q}$ theory would predict that the alveolar gases would equilibrate with venous blood and no gas exchange would occur, whereas the reality is that the $N_2$ in the alveolar spaces diffuses away and the lung oxygenates the blood (the diffusional uptake of $O_2$ across the alveolar membrane generates a convective flow of $O_2$ in the airway). By way of specific example, the inspired gas for patient 2 had a $P_{O2}$ of 606 mmHg. For a $\dot{V}/\dot{Q}$ of 0.05, the $P_{N2}$ calculated for the low $\dot{V}/\dot{Q}$ compartment was 527 mmHg, which compares with a $P_{N2}$ of 103 mmHg calculated for the high $\dot{V}/\dot{Q}$ compartment and a $P_{N2}$ of 107.0 mmHg in the inspired gas. Inevitably, this will generate significant diffusion out of the low $\dot{V}/\dot{Q}$ alveolar space, both back into the airways and also into the blood stream once the $P_{N2}$ of the blood has been lowered sufficiently by its exposure to lower values for $P_{N2}$ associated with other, higher $\dot{V}/\dot{Q}$ compartments. Theoretically, the problem arises because *Eq 1–3* deal only with convection, and assume that the movement of gas by diffusion within the gas phase can be neglected. Under conditions where the $N_2$ concentration gradients in the lung are going to be very large and convection slow, such as predicted in these patients, the use of *Eqs 1–3* is not valid.

Several limitations exist with this study. First, the analysis is limited to the five patients for whom Busana *et al.* were able to collect the necessary data. This number is insufficient to generalise any results to the overall patient population with any degree of certainty. Second, some starting parameters had to be estimated, for example the mixed venous concentrations and partial pressures, from measurements of mixed venous saturation. Third, the absence of any data pertaining to ventilation meant that a useful constraint on possible solutions was also missing.

In conclusion, the $\dot{V}/\dot{Q}$ distributions that arise when the shunt blood flow fraction is assumed proportional to the non-aerated lung fraction are unlikely to represent the true state of gas exchange in these severely ill COVID-19 patients. We feel a more likely interpretation of the derangements of gas exchange in these patients is that the $\dot{V}/\dot{Q}$ distributions are not as extreme as calculated by either Busana *et al.* or ourselves, but rather that the shunt fraction is higher than the fraction of non-aerated lung. One possible explanation may be that infection with Sars-CoV-2 impairs hypoxic pulmonary vasoconstriction (HPV), which normally would cause vasoconstriction within the non-aerated lung, limiting the blood supply to these regions. This impairment of HPV, which may well arise from direct infection and damage of the pulmonary vascular endothelium by Sars-Cov-2 [9, 10], could contribute to a shunt fraction that

is greater than that expected from the amount of non-aerated (consolidated) lung. It seems to us that a simple statistic of interest would be the ratio between the shunt flow fraction calculated for the three-compartment model relative to the fraction of non-aerated lung ($\dot{Q}s_{3CM}/\dot{Q}s_{BM}$). This statistic could be compared with the same statistic calculated for patients suffering from non-COVID-19 ARDS. Busana *et al.* describe such control patients in the appendix to their paper, but there is not sufficient detail to undertake the calculation and compare the results with those for the COVID-19 patients.

## Author Contributions

**Conceptualization:** Peter A. Robbins.

**Data curation:** Haopeng Xu, Nayia Petousi, Peter A. Robbins.

**Visualization:** Haopeng Xu.

**Writing – original draft:** Peter A. Robbins.

**Writing – review & editing:** Haopeng Xu, Nayia Petousi, Peter A. Robbins.

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
