## [Decision Letter · Decision Letter 0]

4 Nov 2021

PONE-D-21-23433Identifying putative ventilation-perfusion distributions in COVID-19 pneumoniaPLOS ONE

Dear Dr. Robbins,

Thank you for submitting your manuscript to PLOS ONE. After careful consideration, we feel that it has merit but does not fully meet PLOS ONE’s publication criteria as it currently stands. Therefore, we invite you to submit a revised version of the manuscript that addresses the points raised during the review process.

We look forward to receiving your revised manuscript.

Kind regards,

Adélia Sequeira, Ph.D

Academic Editor

PLOS ONE

Journal Requirements:

Additional Editor Comments

The manuscript deals with a very interesting topic and a new mathematical model.

Based on the advice received from our reviewers, I feel that your manuscript could be reconsidered for publication should you be prepared to incorporate major revisions. When preparing your revised manuscript, you are asked to carefully consider the reviewers comments, which can be found below, and submit a list of responses to the comments.

One particular issue needs your special attention: a complete rewriting of the abstract and introduction is required in order to understand the main purpose of the paper.

Reviewers' comments:

Reviewer's Responses to Questions

**Comments to the Author**

1. Is the manuscript technically sound, and do the data support the conclusions?

Reviewer #1: No

Reviewer #2: Partly

Reviewer #3: Yes

2. Has the statistical analysis been performed appropriately and rigorously? 

Reviewer #1: No

Reviewer #2: N/A

Reviewer #3: N/A

3. Have the authors made all data underlying the findings in their manuscript fully available?

Reviewer #1: Yes

Reviewer #2: Yes

Reviewer #3: Yes

4. Is the manuscript presented in an intelligible fashion and written in standard English?

Reviewer #1: Yes

Reviewer #2: Yes

Reviewer #3: Yes

5. Review Comments to the Author

Reviewer #1: This paper deals with the identification of ventilation-perfusion imbalances in COVID-19 pneumonia. The topic is interesting. However, the paper presents major drawbacks for which the paper cannot be accepted in the present form: it requires a full rewriting before being reconsidered.

1) One should read well down into the paper, close to the end, to understand the purpose of the paper and its goals. Basically, the paper proposes a modification of a mathematical model that is to be personalized to patients. A complete rewriting of the abstract and introduction is required.

2) Several unclear and misleading terms are used. Accuracy and existence of solutions must be properly accompanied with mathematical and numerical models, other than recast in the proper context. The identification of solutions is unclear too.

3) Model personalization and validation are not mentioned; if so, they are presented in the paper in a narrative manner that prevents comprehension of the procedures used.

4) The mathematical model is not clearly presented. The authors discuss how they arrived to the model, but then the set of equations to be solved (compartmental 0D models) are not clearly written. How are these differing from other models in literature? How is the model validated?

5) The issue of mass conservation is introduced in the abstract, but then the discussion at the end of the paper fail to highlight the reason for which this is unlikely to hold. In addition, one may find hard to believe that a physical principle in classic mechanics like mass conservation does not hold in this context.

6) The motivation to this work appears to come from drawbacks from a paper in literature by Busana et al., which is unpleased by the authors. I believe that, regardless of the pros and cons of the mentioned previous study, negative views of this work should be recast in a more constructive manner.

Specific comments for the abstract.

7) Abstract, 21 (and Introduction). Rephrase the first part of the abstract. Motivations to this paper and its content should not move from negative views on previously appeared papers. “For no patient they did obtain accurate results”. Rather, try recasting the previous work in a positive manner.

8) Abstract, 27. After accuracy of previous results is discussed, the goal of the paper is stated “to determine whether such solutions exist, and if so, to develop accurate method by which possible solutions can be identified”. What is the the analysis of solutions’ existence that is established in this paper? What does solution identification mean?

9) Abstract, 30. What does it mean that “no solution was possible”?

10) Abstract, 32. What does it mean “precise solution to the problem”?

11) Abstract, 34. In which sense are solutions “exact”?

12) Abstract, 34-35. The statement is unclear. What is the purpose of the method that is failing? Performing numerical discretization of the mathematical model? Or performing data assimilation?

13) Abstract, 36. What is the conclusions and the proposed remedy to the unlikely assumption made on mass conservation?

Reviewer #2: The paper deals with a new mathematical model to determine a ventilation-perfusion distribution able to reproduce the oxygen and Co2 partial pressure in COVID-19 patients. The authors start from a previous model and improve it, showing with their results applied to a cohort of 5 patients the improvements with respect to the previous results.

The topic of the paper is of great interest due to an attempt to better understand COVID-19 effects on ventilation and perfusion of lungs in a quantitative way. However, some remarks are due and the authors should carefully fix the following points

MAJOR REMARKS:

1. The authors refer to the work of Busana et al in terms too competitive, providing definite statements on such a work, e.g.

“… they subsequently failed to find any accurate solutions for potential …”

“… result from the hypothesis of Busana et al. are unlikely to represent the true state of gas exchange in these severely ill COVID-19 patients.”

The authors should clearly state that their model is an extension of that of Busana et al, with an improvement in some sense of the results, acknowledging the fact that the Busana model is the starting point.

2. Related to the previous point, the authors used too much strong statements about their results, e.g.:

“… we have demonstrated that no such distributions exist for their first …”

“… we have demonstrated that multiple, indeed infinite sets, of potential V̇/Q̇ distributions exist…”

It seems to me that the results of this paper are more reasonable than the ones found in Busana et al, but that there is no any validation, so I would avoid to use “demonstrate”.

3. Related to the previous point: The authors used the available data for the parameter calibration, but not for the validation, is this true? If yes, this should be clearly stated in the text

4. Also the number of cases (5) is not enough to demonstrate anything. It is noticeable to have some data after 1 year from pandemic, but the authors should again change the tone of their sentences, without any definite answer. Their results in fact “seem to show that …. “

Moreover, if I well understood, the data were obtained by Busana et al and this is another reason why the authors should refer to this paper in different terms (see point 1)

4. Methods – Overview: The journal is read by scientists with different expertise, thus I suggest to better contextualize the physical processes and the method. For example: What are the compartments? What is the pure shunt and pure deadspace? Not all the readers are familiar with this.

5. Eqns (1) and (2): I am not sure that the authors explicitly give the expression of g. In any case, Eqn (2) is useless, is the same of (1)

6. Methods: the authors should summarize all the procedure with a final algorithm or better with a flowchart, highlighting:

- the input

- how could they be obtained (measures, assumptions, …)

- the output

- their clinical relevance

7. Figures 1 and 2: It seems to me that different behaviours are experienced by Patients 2,3 vs 4,5. Please comment on this

8. Details on the numerical methods used to find solutions should be provided

9. The clinical relevance of the results should be discussed

10. Limitations and future perspective should be added

MINOR REMARKS:

1. Line 294 should be after the caption

Reviewer #3: n.a (this is not a review of the manuscript)

n.a (this is not a review of the manuscript)

n.a (this is not a review of the manuscript)

n.a (this is not a review of the manuscript)

n.a (this is not a review of the manuscript)

6. PLOS authors have the option to publish the peer review history of their article (what does this mean?). If published, this will include your full peer review and any attached files.

Reviewer #1: No

Reviewer #2: No

Reviewer #3: No

---

## [Author Response · Author response to Decision Letter 0]

17 Dec 2021

Reviewer #1

This paper deals with the identification of ventilation-perfusion imbalances in COVID-19 pneumonia. The topic is interesting. However, the paper presents major drawbacks for which the paper cannot be accepted in the present form: it requires a full rewriting before being reconsidered.

Comment 1: One should read well down into the paper, close to the end, to understand the purpose of the paper and its goals. Basically, the paper proposes a modification of a mathematical model that is to be personalized to patients. A complete rewriting of the abstract and introduction is required.

Response to comment 1: We have rewritten the abstract and introduction following this comment. We hope we have clarified the purpose of the paper much earlier in the text. 

Comment 2: Several unclear and misleading terms are used. Accuracy and existence of solutions must be properly accompanied with mathematical and numerical models, other than recast in the proper context. The identification of solutions is unclear too.

Response to comment 2: We have now specified what is meant by accuracy and existence of solutions so as to make these terms clear. 

Comment 3: Model personalization and validation are not mentioned; if so, they are presented in the paper in a narrative manner that prevents comprehension of the procedures used.

Response to comment 3: The general structure of the compartmental model is the same for each patient, but the parameters distributing blood flow and ventilation are personalized for each patient. We agree that this was not fully clear in the original manuscript and have now re-structured the Methods to separate the governing equations of the compartmental model from the personalization of the distributions. 

Comment 4: The mathematical model is not clearly presented. The authors discuss how they arrived to the model, but then the set of equations to be solved (compartmental 0D models) are not clearly written. How are these differing from other models in literature? How is the model validated?

Response to comment 4: This is fair comment, and we have restructured the Methods to clarify this. The underlying compartmental model, based on mass balance, is longstanding, widely accepted and has been used by many investigators. Apart from details such as the number of compartments and the detailed form of the blood-gas dissociation curves, the model we used was essentially the same as that of Busana et al. The difference between the two papers is essentially how you personalize the distributions for individual patients.

The issues around validation in this area are much harder. We can show that a distribution is a candidate distribution for an individual, in the sense that it reproduces the patient’s arterial PCO2 and PO2. In general, however, there is always an infinite number of candidate distributions, whereas there can only be one physical distribution pertaining in the lungs. What does tend to be common amongst a set of candidate distributions are certain qualitative features, such as their overall width or the number of their peaks. Common between Busana et al and our manuscript are the extraordinarily wide distributions seen with the four patients in whom there are solutions. 

Comment 5: The issue of mass conservation is introduced in the abstract, but then the discussion at the end of the paper fail to highlight the reason for which this is unlikely to hold. In addition, one may find hard to believe that a physical principle in classic mechanics like mass conservation does not hold in this context.

Response to comment 5: Sorry. This has been misunderstood. Our point was actually the other way around, namely a result that violates a fundamental principle of classic mechanics means there is something wrong with the assumptions that generated it (NOT the assumptions of classical mechanics), here the assumption of the shunt fraction value. We have re-written this point. 

Comment 6: The motivation to this work appears to come from drawbacks from a paper in literature by Busana et al., which is unpleased by the authors. I believe that, regardless of the pros and cons of the mentioned previous study, negative views of this work should be recast in a more constructive manner.

Response to Comment 6: Thank you. We are glad that you flagged this because it was not intended. Busana et al have obtained patient data during the pandemic in a manner that has been remarkably hard to do. They should be commended for it. We have checked and, if necessary, rephrased our references to the Busana et al paper in the manuscript. 

Comment 7: Abstract, 21 (and Introduction). Rephrase the first part of the abstract. Motivations to this paper and its content should not move from negative views on previously appeared papers. “For no patient they did obtain accurate results”. Rather, try recasting the previous work in a positive manner.

Response to Comment 7: Agreed. We have re-written this.

Comment 8: Abstract, 27. After accuracy of previous results is discussed, the goal of the paper is stated “to determine whether such solutions exist, and if so, to develop accurate method by which possible solutions can be identified”. What is the the analysis of solutions’ existence that is established in this paper? What does solution identification mean?

Response to Comment 8: Busana et al have assumed that the shunt flow fraction was proportional to the non-aerated lung fruction, and calculated this from CT imaging. Using this assumed shunt flow fraction as given by Busana et al (and the non-shunt fraction calculated from this by subtracting from cardiac output), we used the three-compartment model to investigate whether there exist V̇/Q̇ distributions for the non-shunt part of the lung that can produce the measured arterial PCO2 and PO2. (this is what we mean by solutions’ existence). This model’s theoretical importance is that for any real V̇/Q̇ distribution, no matter how complex, there always exists a corresponding three-compartment model that can exactly replicate the patient’s PaCO2 and PaO2. Using the well established three-compartment model one can calculate the maximum physiologically possible shunt fraction (Q̇s3CM) and the minimum physiologically possible non-shunt fraction (Q̇ns3CM) for each patient. For patient 1, the assumption that shunt flow fraction was proportional to the non-aerated lung fruction cannot hold, as this non-shunt flow fraction by Busana et al (Q̇nsBM) is too low, lower than the minimum possible (Q̇ns3CM), to be able to support gas-exchange to produce the patient’s PCO2 and PO2: so no solution exists (no V/Q distribution) with this assumption. For the other patients, solutions are possible with this assumed shunt and non-shunt fraction (derived from imaging), but we wanted to find a method that provides more accurate solutions, i.e. V/Q distributions that produce arterial PCO2 and PO2 that are within the measurement error of the blood gas analyser (which we take as a maximum error of 1 mmHg) (this is what is meant by solution identification).

Comments 9, 10 and 11: Abstract, 30. What does it mean that “no solution was possible”; Abstract, 32. What does it mean “precise solution to the problem”? Abstract, 34. In which sense are solutions “exact”?

Response to Comments 9, 10 and 11: We have revised the abstract to address these issues. The meanings were:. 

No solution was possible: no parameterization of the model of V/Q distribution exists that can reproduce the arterial PCO2, and PO2. (This occurs for patient 1 where the specified total blood flow is simply too small to deliver the required gas exchange.) 

Precision of the solution: defined in terms of the residuals for the arterial PCO2 and PO2. This is the difference between the patient’s measured data and the predicted values from the model. We aim for a precision that is within that reasonably associated with the blood gas analyser. 

Exact solution: we have removed this term because all solutions require a degree of numerical estimation. 

Comment 12: Abstract, 34-35. The statement is unclear. What is the purpose of the method that is failing? Performing numerical discretization of the mathematical model? Or performing data assimilation?

Response to Comment 12: The classical equations referred to here model convective transport of gases to and from well mixed compartments, and assume that diffusion can be neglected (apart from equilibration across the alveolar membrane). This assumption is reasonable under most conditions. However, the high inspired PO2 in these patients coupled with the extraordinarily wide V/Q ratios will create certain regions of the lung where convective tramsport in the gas phase is very slow, but at the same time where very large concentration gradients are predicted to persist. Under these conditions, we think the assumption that diffusion can be neglected breaks down and the model is no longer valid. 

Comment 13: Abstract, 36. What is the conclusions and the proposed remedy to the unlikely assumption made on mass conservation?

Response to Comment 13: See response to comment 5. Fundamentally, we think the shunt flow assumption is not sufficiently accurate. 

Reviewer #2

Comment 1: The authors refer to the work of Busana et al in terms too competitive, providing definite statements on such a work, e.g.

“… they subsequently failed to find any accurate solutions for potential …”

“… result from the hypothesis of Busana et al. are unlikely to represent the true state of gas exchange in these severely ill COVID-19 patients.”

The authors should clearly state that their model is an extension of that of Busana et al, with an improvement in some sense of the results, acknowledging the fact that the Busana model is the starting point.

Response to comment 1: Sorry. We are glad the reviewer makes this point as it really wasn’t intended. Busana et al should be congratulated for managing to obtain these data during the course of the pandemic. We have checked the wording and changed where appropriate. We have also clarified that what we set out to achieve was an alternative approach to Busana et al’s multiple simulation approach that could provide a direct calculation of parameterisations that should reproduce a patient’s arterial PCO2 and PO2. 

Comment 2. Related to the previous point, the authors used too much strong statements about their results, e.g.:

“… we have demonstrated that no such distributions exist for their first …”

“… we have demonstrated that multiple, indeed infinite sets, of potential V̇/Q̇ distributions exist…”

It seems to me that the results of this paper are more reasonable than the ones found in Busana et al, but that there is no any validation, so I would avoid to use “demonstrate”.

Response to comment 2: We don’t completely agree with the reviewer here.

Busana et al found no possible solutions for their first patient. We use the three-compartment model to show that the postulated blood flow to the aerated lung is simply too low to support the gas exchange required – there is no solution to the problem (and therefore the starting assumptions concerning shut flow have to be wrong).

For the second statement, the key word is ‘potential’ (or candidate) distributions because they do reproduce the patient’s arterial PCO2 and PO2. However, we completely agree with the reviewer re: validation and have adjusted the text accordingly. 

Comment 3. Related to the previous point: The authors used the available data for the parameter calibration, but not for the validation, is this true? If yes, this should be clearly stated in the text

Response to comment 3: Yes, that is completely correct. Furthermore, we argue that the extraordinarily-wide distributions are unlikely to exist in these patients given their very high inspired PO2. Thus, if there were some way of validating these results experimentally, we would expect the validation to fail. We think the problem resides with the ‘shunt flow assumption’, as covered in the Discussion. 

Comment 4. Also the number of cases (5) is not enough to demonstrate anything. It is noticeable to have some data after 1 year from pandemic, but the authors should again change the tone of their sentences, without any definite answer. Their results in fact “seem to show that …. “

Moreover, if I well understood, the data were obtained by Busana et al and this is another reason why the authors should refer to this paper in different terms (see point 1)

Response to comment 4: Whilst we can demonstrate things as they apply to individual patients, we agree with the reviewer that 5 patients are too few from which to draw generalized conclusions in relation to the patient population as a whole. We have checked the manuscript to ensure any generalizations have that caveat clearly stated. We also agree with the second point (see our response to point 1).

Comment 5. Methods – Overview: The journal is read by scientists with different expertise, thus I suggest to better contextualize the physical processes and the method. For example: What are the compartments? What is the pure shunt and pure deadspace? Not all the readers are familiar with this.

Response to comment 5: This is a very fair comment. We have revised the manuscript to include those definitions, but also more generally, we now start the methods with a description of the compartmental model on which this study is based.

Comment 6. Eqns (1) and (2): I am not sure that the authors explicitly give the expression of g. In any case, Eqn (2) is useless, is the same of (1)

Response to comment 6: No we don’t give an explicit expression for g. In this study, we use quite a detailed mathematical model of the blood, which involves the simultaneous solution of four non-linear equations for the plasma H+ concentration, the intra-erythrocytic H+ concentration, the permeant strong ion difference in the plasma and the permeant strong ion difference in the red cells. These values are then substituted into other equations to get the blood CO2 and O2 contents. If we try to set this all out here (apart from anything else, there are 50-100 chemical constants), it will serve to confuse rather than help. It is better just to give the reference here (O’Neil at al) so that the interested reader can follow it up. Other readers only need to know what it is doing in order to follow the paper.

We agree equation 2 was probably unnecessary – the only points we were trying to make were that the inverse function exists (i.e. the function is both injective and surjective) and that it is possible to use it in calculations. 

Comment 7. Methods: the authors should summarize all the procedure with a final algorithm or better with a flowchart, highlighting:

- the input

- how could they be obtained (measures, assumptions, …)

- the output

- their clinical relevance

Response to comment 7: This is an excellent idea and we have now introduced a flowchart into the Methods section. 

Comment 8. Figures 1 and 2: It seems to me that different behaviours are experienced by Patients 2,3 vs 4,5. Please comment on this.

Response to comment 8: There are indeed some quantitative differences, and of course the patients differ in their arterial blood gases and the level of inspired O2 that they are receiving (table 1). However, qualitatively the results are similar. They are all monotonic increasing functions and the solutions are all confined to exceptionally low values for V/Q. 

Comment 9. Details on the numerical methods used to find solutions should be provided

Response to comment 9: We used the in-built equation solvers in Matlab (there is nothing particularly tricky about the solutions). We have added this point to the methods and referred to the particular solvers that were employed. 

Comment 10. The clinical relevance of the results should be discussed.

Response to comment 10: We have added a section in the Discussion about the results’ clinical relevance.

Comment 11. Limitations and future perspective should be added

Response to comment 11: A section on limitations and future perspective has been added in the Discussion. 

Comment 12. Line 294 should be after the caption.

Response to comment 12: Done.

---

## [Decision Letter · Decision Letter 1]

17 Feb 2022

PONE-D-21-23433R1Identifying putative ventilation-perfusion distributions in COVID-19 pneumoniaPLOS ONE

Dear Dr. Robbins,

Thank you for submitting your manuscript to PLOS ONE. After careful consideration, we feel that it has merit but does not fully meet PLOS ONE’s publication criteria as it currently stands. Therefore, we invite you to submit a revised version of the manuscript that addresses the points raised during the review process.

We look forward to receiving your revised manuscript.

Kind regards,

António M. Lopes, PhD

Academic Editor

PLOS ONE

Reviewers' comments:

Reviewer's Responses to Questions

**Comments to the Author**

1. If the authors have adequately addressed your comments raised in a previous round of review and you feel that this manuscript is now acceptable for publication, you may indicate that here to bypass the “Comments to the Author” section, enter your conflict of interest statement in the “Confidential to Editor” section, and submit your "Accept" recommendation.

Reviewer #1: (No Response)

Reviewer #2: (No Response)

2. Is the manuscript technically sound, and do the data support the conclusions?

Reviewer #1: Partly

Reviewer #2: Partly

3. Has the statistical analysis been performed appropriately and rigorously? 

Reviewer #1: No

Reviewer #2: N/A

4. Have the authors made all data underlying the findings in their manuscript fully available?

Reviewer #1: Yes

Reviewer #2: No

5. Is the manuscript presented in an intelligible fashion and written in standard English?

Reviewer #1: Yes

Reviewer #2: Yes

6. Review Comments to the Author

Reviewer #1: The paper improved after a significant revision that partially incorporated the major and minor questions. I think the paper has merit, even if some major points persist and must be thoroughly addressed.

I am very confused by the use of the words “solution”, “model”, and “problem”. Mathematically speaking, a solution is a value, number, function, etc… that fulfill the conditions set by equations. Here, I find very hard to understand which are the equations to be solved (models and problems) and which variables are the solutions by reading through the text. Figure 1 is adding confusion rather than giving a clear picture of the problems/equations and solutions searched for. What is meaning that a “solution is not possible”? I also find very difficult to identify data and solutions (?) in the models.

The concept of “distribution” should be introduced and carefully presented. If I understand correctly, this work revolves around the idea of calculating such distribution instead of a sort of “trial and error” approach by Busana et al., who explore the space of plausible parameters to identify a plausible combination of these ones. Can this distribution interpreted instead as a combination of values?

118: “conservation of matter”. Shouldn’t be conservation of mass instead? Here, we are not working in the framework of relativistic mechanics, but of classical mechanics.

150: Eq. (5). g is not defined, if not much later in the text. Similarly for other mathematical notation.

Equations are numbered in round brackets (XY). However, in the text they are referred to as Eq XY. Instead, bibliographic references are cited in round brackets (in place of more common squared brackets), which is adding confusion to the “Method” section.

356-365: are these part of the table caption?

Reviewer #2: It is hard to evaluate the changes made by the authors. They should clearly indicate in their answers the number of pages and lines where changes have been made and, possibly, indicate in colours such changes in the text

7. PLOS authors have the option to publish the peer review history of their article (what does this mean?). If published, this will include your full peer review and any attached files.

Reviewer #1: No

Reviewer #2: No

---

## [Author Response · Author response to Decision Letter 1]

9 Mar 2022

Submitted title

Identifying putative ventilation-perfusion distributions in COVID-19 pneumonia

Respond to reviewers’ comments

Reviewer #1

The paper improved after a significant revision that partially incorporated the major and minor questions. I think the paper has merit, even if some major points persist and must be thoroughly addressed.

Comment 1 

I am very confused by the use of the words “solution”, “model”, and “problem”. Mathematically speaking, a solution is a value, number, function, etc… that fulfill the conditions set by equations. Here, I find very hard to understand which are the equations to be solved (models and problems) and which variables are the solutions by reading through the text. Figure 1 is adding confusion rather than giving a clear picture of the problems/equations and solutions searched for. What is meaning that a “solution is not possible”? I also find very difficult to identify data and solutions (?) in the models.

Response to comment 1

Thank you – this is very helpful comment to guide further revision. We have added a paragraph at the beginning of the Methods to define the terms ‘solution’, ‘model’ and ‘problem’.

A solution is a set of paired values for ventilation and perfusion {Vi, Qi}, that will result in calculated values for arterial PCO2 and PO2 that, to within experimental error, match the measured values from the patient. We illustrate solutions in the figures. The meaning of a ‘solution is not possible’ is that, after applying constraints relating to the patient, there are no sets {Vi, ,Qi} that form a solution to the problem. This can arise if one or more of the constraints is wrong.

Comment 2

The concept of “distribution” should be introduced and carefully presented. If I understand correctly, this work revolves around the idea of calculating such distribution instead of a sort of “trial and error” approach by Busana et al., who explore the space of plausible parameters to identify a plausible combination of these ones. Can this distribution interpreted instead as a combination of values?

Response to comment 2

We have added a definition of distribution in the paragraph at the start of the Methods. Busana et al do not state what their parameters are, but basically their parameters are used to generate the ‘ V/Q distribution’, which is the set of paired values {Vi, Qi}. The number of paired values in this set can be varied. We explore the three-compartment model of the lung – which has three pairs of values in this set (and has a very important place in the development of theory around gas exchange in the lung), a four-compartment model of the lung, and a ‘multi’-compartment model of the lungs, where the number of Vi, Qi pairs is very much higher and where it is really being used as a computational approximation to a continuous distribution (where the pairs of values Vi, Qi in the set {Vi, Qi} would be infinite).

So the answer to your question is yes. The distribution is the set {Vi, Qi}.

Comment 3

118: “conservation of matter”. Shouldn’t be conservation of mass instead? Here, we are not working in the framework of relativistic mechanics, but of classical mechanics.

Response to comment 3

We have changed this.

Comment 4: 150: Eq. (5). g is not defined, if not much later in the text. Similarly for other mathematical notation.

Response to comment 4

As indicated in the preceding paragraph, g is a function that represents the blood gas model of reference 7. To make this clearer, we have now added the reference in the sentence immediately preceding equation 5. 

Comment 5

Equations are numbered in round brackets (XY). However, in the text they are referred to as Eq XY. Instead, bibliographic references are cited in round brackets (in place of more common squared brackets), which is adding confusion to the “Method” section.

Response to comment 5

We have revised the manuscript so that the citations now appear in the text in squared brackets.

Comment 6

356-365: are these part of the table caption?

Response to comment 6

Yes, that is correct. 

Reviewer #2

It is hard to evaluate the changes made by the authors. They should clearly indicate in their answers the number of pages and lines where changes have been made and, possibly, indicate in colours such changes in the text.

Response to reviewer #2

Our apologies for this. We used the track changes feature to highlight all the text that had changed, but there was so much of it, and so much editing, that the end result was a bit of a mess. The changes are much clearer in this second revision.

---

## [Decision Letter · Decision Letter 2]

11 May 2022

PONE-D-21-23433R2

Identifying putative ventilation-perfusion distributions in COVID-19 pneumonia

PLOS ONE

Dear Dr. Robbins,

Thank you for submitting your manuscript to PLOS ONE. After careful consideration, we feel that it has merit but does not fully meet PLOS ONE’s publication criteria as it currently stands. Therefore, we invite you to submit a revised version of the manuscript that addresses the points raised during the review process.

*You may see that reviewer 2 is still unclear regarding the changes made to the mansucript as a result of their comments provided in the decision letter dated November 4^th^ 2021. We appreciate that while in the rebuttal letter following the first round of review, you have provided a response to every comment made by this reviewer, however it is not clear the changes made to the mansucript text as a result of their feedback.During this round of revision, we suggest explicitly referring to what as changed in the revised version of the mansucript when compared to the original submitted mansucript (using line numbers if possible) to support your responses to the reviewers 2 comments. This will help the reviewers and Academic Editor in re-evaluating your mansucript.*

We look forward to receiving your revised manuscript.

Kind regards,

Lucinda Shen 

Staff Editor 

on behalf of 

António M. Lopes, PhD

Academic Editor

PLOS ONE

**Additional Editor Comments:**

The authors should pay attention to the comments of reviewer #2.

Reviewers' comments:

Reviewer's Responses to Questions

**Comments to the Author**

1. If the authors have adequately addressed your comments raised in a previous round of review and you feel that this manuscript is now acceptable for publication, you may indicate that here to bypass the “Comments to the Author” section, enter your conflict of interest statement in the “Confidential to Editor” section, and submit your "Accept" recommendation.

Reviewer #1: (No Response)

Reviewer #2: (No Response)

2. Is the manuscript technically sound, and do the data support the conclusions?

Reviewer #1: Partly

Reviewer #2: No

3. Has the statistical analysis been performed appropriately and rigorously? 

Reviewer #1: N/A

Reviewer #2: N/A

4. Have the authors made all data underlying the findings in their manuscript fully available?

Reviewer #1: (No Response)

Reviewer #2: No

5. Is the manuscript presented in an intelligible fashion and written in standard English?

Reviewer #1: (No Response)

Reviewer #2: Yes

6. Review Comments to the Author

Reviewer #1: Response to comment 1

Thank you – this is very helpful comment to guide further revision. We have added a paragraph at the beginning of the Methods to define the terms ‘solution’, ‘model’ and ‘problem’. A solution is a set of paired values for ventilation and perfusion {Vi, Qi}, that will result in calculated values for arterial PCO2 and PO2 that, to within experimental error, match the measured values from the patient. We illustrate solutions in the figures. The meaning of a ‘solution is not possible’ is that, after applying constraints relating to the patient, there are Response to Reviewers no sets {Vi, ,Qi} that form a solution to the problem. This can arise if one or more of the constraints is wrong.

>>> This must be made very clear from the beginning, that is from the abstract.

Response to comment 2

We have added a definition of distribution in the paragraph at the start of the Methods. Busana et al do not state what their parameters are, but basically their parameters are used to generate the ‘ V/Q distribution’, which is the set of paired values {Vi, Qi}. The number of paired values in this set can be varied. We explore the three-compartment model of the lung – which has three pairs of values in this set (and has a very important place in the development of theory around gas exchange in the lung), a four-compartment model of the lung, and a ‘multi’-compartment model of the lungs, where the number of Vi, Qi pairs is very much higher and where it is really being used as a computational approximation to a continuous distribution (where the pairs of values Vi, Qi in the set {Vi, Qi} would be infinite). So the answer to your question is yes. The distribution is the set {Vi, Qi}.

>>> Also this must be very clear from the beginning of the paper.

>>> I recommend to further ease negative views on the paper of Busana et al. (like “failed to find any solution”, abstract line 29)

Reviewer #2: I cannot find the answers to my remarks. IN previous submission, the changes in the text were missing, now the answers are missing. It is difficult to evaluate the improvement of the paper

7. PLOS authors have the option to publish the peer review history of their article (what does this mean?). If published, this will include your full peer review and any attached files.

Reviewer #1: No

Reviewer #2: No

---

## [Author Response · Author response to Decision Letter 2]

7 Jul 2022

RESPONSE TO REVIEWERS’ COMMENTS (3rd REVISION) 

Reviewer #1

Comment 1

‘Thank you – this is very helpful comment to guide further revision……’

>>> This must be made very clear from the beginning, that is from the abstract.

Response to comment 1

We have now included these points in the abstract to the extent that it is possible while still remaining within the tight word limit required.

Change R2 Lines 23 - 26 ‘They assumed that the shunt flow fraction was proportional to the non-aerated lung fraction, and, by randomly generating 106 different bimodal distributions for blood flow, sought to identify ventilation-perfusion (V̇/Q̇) distributions for the aerated element of the lung that would generate the observed arterial partial pressures of CO2 and O2 (PaCO2 and PaO2¬). ’

To R3 Lines 22 - 26 ‘They assumed that shunt flow fraction was proportional to the non-aerated lung fraction, and, by randomly generating 106 different bimodal distributions for the ventilation-perfusion (V̇/Q̇) ratios in the lung, specified as sets of paired values {V̇i, Q̇i}, sought to identify as solutions those that generated the observed arterial partial pressures of CO2 and O2 (PaCO2 and PaO2¬).’

Comment 2

‘We have added a definition of distribution in the paragraph……’ 

>>> Also this must be very clear from the beginning of the paper.

Response to comment 2

We have now included these points within the introduction to the paper.

Insert R3 Lines 57 - 59 ‘Any particular V̇/Q̇ distribution is then specified as a set of paired values for ventilation and perfusion {V̇i, Q̇i}, where i is the index for the compartment.

Comment 3

>>> I recommend to further ease negative views on the paper of Busana et al. (like “failed to find any solution”, abstract line 29)

Response to comment 3

We have removed the word ‘failed’ in case it is seen as pejorative.

Change R2 Line 29 ‘For the one patient in whom Busana et al. failed’ 

To R3 Line 29 ‘For the one patient in whom Busana et al. did not find solutions’.

Reviewer #2

Comment 1

I cannot find the answers to my remarks. IN previous submission, the changes in the text were missing, now the answers are missing. It is difficult to evaluate the improvement of the paper

Response to comment 1

We appreciate that our use of the ‘track changes’ feature of Word to identify where changes had been made was not very successful. For this revision, we provide a compilation of all the reviewers’ (both reviewer 1 and reviewer 2) prior comments at each stage of the review process, together with all of our responses. Within this compilation, we have inserted, for each comment, a reference in red to the line numbers of the original manuscript where changes were made, together with a reference in red the line numbers where the changes are now to be found in the current (R3) version of the manuscript. We hope this is helpful.

RESPONSE TO REVIEWERS (1st REVISION)

Reviewer #1

This paper deals with the identification of ventilation-perfusion imbalances in COVID-19 pneumonia. The topic is interesting. However, the paper presents major drawbacks for which the paper cannot be accepted in the present form: it requires a full rewriting before being reconsidered.

Comment 1

One should read well down into the paper, close to the end, to understand the purpose of the paper and its goals. Basically, the paper proposes a modification of a mathematical model that is to be personalized to patients. A complete rewriting of the abstract and introduction is required.

Response to comment 1

We have rewritten the abstract and introduction following this comment. We hope we have clarified the purpose of the paper much earlier in the text. 

Remove Original Lines 20 - 38 Old Abstract

Replace with R3 Lines 20 - 37 New Abstract.

Remove Original Lines 40 - 73 Old Introduction

Replace with R3 Lines 39 - 87 New Introduction.

Comment 2 

Several unclear and misleading terms are used. Accuracy and existence of solutions must be properly accompanied with mathematical and numerical models, other than recast in the proper context. The identification of solutions is unclear too.

Response to comment 2

We have now specified what is meant by accuracy and existence of solutions so as to make these terms clear. 

Insert R3 Lines 111 - 113 ‘The accuracy we sought was for the model values for PaCO2 and PaO2 to be within 1 mmHg of the patient’s measured values, although in practice the errors were less that 1x10-2 mmHg or even 1x10-3 mmHg’.

Insert R3 Lines 260 - 263 ‘From this, it follows immediately that, if Q̇ns3CM > Q̇nsBM, then Q̇nsBM is too low to support the gas-exchange required to produce the patient’s measured PaO2 and PaCO2. This provides a basis for our test of whether V̇/Q̇ distributions exist for the shunt flows proposed by Busana et al.’.

Comment 3

Model personalization and validation are not mentioned; if so, they are presented in the paper in a narrative manner that prevents comprehension of the procedures used.

Response to comment 3

The general structure of the compartmental model is the same for each patient, but the parameters distributing blood flow and ventilation are personalized for each patient. We agree that this was not fully clear in the original manuscript and have now re-structured the Methods to separate the governing equations of the compartmental model from the personalization of the distributions. 

Insert R3 Lines 128 - 166 New sections to separate governing equations from personalization of distributions.

Comment 4: The mathematical model is not clearly presented. The authors discuss how they arrived to the model, but then the set of equations to be solved (compartmental 0D models) are not clearly written. How are these differing from other models in literature? How is the model validated?

Response to comment 4

This is fair comment, and we have restructured the Methods to clarify this. The underlying compartmental model, based on mass balance, is longstanding, widely accepted and has been used by many investigators. Apart from details such as the number of compartments and the detailed form of the blood-gas dissociation curves, the model we used was essentially the same as that of Busana et al. The difference between the two papers is essentially how you personalize the distributions for individual patients.

The issues around validation in this area are much harder. We can show that a distribution is a candidate distribution for an individual, in the sense that it reproduces the patient’s arterial PCO2 and PO2. In general, however, there is always an infinite number of candidate distributions, whereas there can only be one physical distribution pertaining in the lungs. What does tend to be common amongst a set of candidate distributions are certain qualitative features, such as their overall width or the number of their peaks. Common between Busana et al and our manuscript are the extraordinarily wide distributions seen with the four patients in whom there are solutions. 

Removed Original Lines 76 - 117 Old section of Overview of the method

Insert R3 Lines 90 - 126 New section of Overview to clarify the problems in the paper and steps of solving them. 

Comment 5

The issue of mass conservation is introduced in the abstract, but then the discussion at the end of the paper fail to highlight the reason for which this is unlikely to hold. In addition, one may find hard to believe that a physical principle in classic mechanics like mass conservation does not hold in this context.

Response to comment 5

Sorry. This has been misunderstood. Our point was actually the other way around, namely a result that violates a fundamental principle of classic mechanics means there is something wrong with the assumptions that generated it (NOT the assumptions of classical mechanics), here the assumption of the shunt fraction value. We have re-written this point. 

Change Original Lines 35 - 38 ‘Finally, we note that an assumption of the classical V̇/Q̇ mass balance equations is unlikely to hold at very low V̇/Q̇ under conditions of high inspired oxygen, and therefore these distributions are an unlikely cause of the observed blood gas values in the patients.’

To R3 Lines 35 - 37 ‘We consider that these wide distributions arise because the assumed value for shunt flow is too low in these patients, and we discuss possible reasons why the assumption relating to shunt flow fraction may break down in COVID-19 pneumonia. ’

Comment 6

The motivation to this work appears to come from drawbacks from a paper in literature by Busana et al., which is unpleased by the authors. I believe that, regardless of the pros and cons of the mentioned previous study, negative views of this work should be recast in a more constructive manner.

Response to Comment 6

Thank you. We are glad that you flagged this because it was not intended. Busana et al have obtained patient data during the pandemic in a manner that has been remarkably hard to do. They should be commended for it. We have checked and, if necessary, rephrased our references to the Busana et al paper in the manuscript. 

Remove Original Lines 20 - 38 Old Abstract

Replace with R3 Lines 20 - 37 New Abstract.

Remove Original Lines 40 - 73 Old Introduction

Replace with R3 Lines 39 - 87 New Introduction.

Comment 7

Abstract, 21 (and Introduction). Rephrase the first part of the abstract. Motivations to this paper and its content should not move from negative views on previously appeared papers. “For no patient they did obtain accurate results”. Rather, try recasting the previous work in a positive manner.

Response to Comment 7

Agreed. We have re-written this.

Remove Orignial Line 25 - 26 ‘For no patient did they obtain an accurate solution.’ 

Change Original lines 34 - 35 ‘We conclude that the absence of accurate solutions in Busana et al. arose through a failure of their method.’ 

To R3 Lines 33 - 35 ‘These distributions were extremely wide and unlikely to be physically realisable, because they predict the maintenance of very large concentration gradients in regions of the lung where convection is slow.’

Comment 8

Abstract, 27. After accuracy of previous results is discussed, the goal of the paper is stated “to determine whether such solutions exist, and if so, to develop accurate method by which possible solutions can be identified”. What is the the analysis of solutions’ existence that is established in this paper? What does solution identification mean?

Response to Comment 8

Busana et al have assumed that the shunt flow fraction was proportional to the non-aerated lung fruction, and calculated this from CT imaging. Using this assumed shunt flow fraction as given by Busana et al (and the non-shunt fraction calculated from this by subtracting from cardiac output), we used the three-compartment model to investigate whether there exist V̇/Q̇ distributions for the non-shunt part of the lung that can produce the measured arterial PCO2 and PO2. (this is what we mean by solutions’ existence). This model’s theoretical importance is that for any real V̇/Q̇ distribution, no matter how complex, there always exists a corresponding three-compartment model that can exactly replicate the patient’s PaCO2 and PaO2. Using the well established three-compartment model one can calculate the maximum physiologically possible shunt fraction (Q̇s3CM) and the minimum physiologically possible non-shunt fraction (Q̇ns3CM) for each patient. For patient 1, uibthe assumption that shunt flow fraction was proportional to the non-aerated lung fruction cannot hold, as this non-shunt flow fraction by Busana et al (Q̇nsBM) is too low, lower than the minimum possible (Q̇ns3CM), to be able to support gas-exchange to produce the patient’s PCO2 and PO2: so no solution exists (no V/Q distribution) with this assumption. For the other patients, solutions are possible with this assumed shunt and non-shunt fraction (derived from imaging), but we wanted to find a method that provides more accurate solutions, i.e. V/Q distributions that produce arterial PCO2 and PO2 that are within the measurement error of the blood gas analyser (which we take as a maximum error of 1 mmHg) (this is what is meant by solution identification).

Comments 9, 10 and 11

Abstract, 30. What does it mean that “no solution was possible”; Abstract, 32. What does it mean “precise solution to the problem”? Abstract, 34. In which sense are solutions “exact”?

Response to Comments 9, 10 and 11

We have revised the abstract to address these issues. The meanings were:

No solution was possible: no parameterization of the model of V/Q distribution exists that can reproduce the arterial PCO2, and PO2. (This occurs for patient 1 where the specified total blood flow is simply too small to deliver the required gas exchange.) 

Change Original Lines 30 - 31 ‘For one of the five patients, we demonstrated that no solution was possible.’

To R3 Line 29 - 31 ‘For the one patient in whom Busana et al. did not find solutions, we demonstrated that the assumed shunt flow fraction led to a non-shunt blood flow that was too low to support the required gas exchange.’

Precision of the solution: defined in terms of the residuals for the arterial PCO2 and PO2. This is the difference between the patient’s measured data and the predicted values from the model. We aim for a precision that is within that reasonably associated with the blood gas analyser. 

Insert R3 Lines 31 - 33 ‘For the other four patients, we found precise solutions (prediction error < 1x10-3 mmHg for both PaCO2 and PaO2), with distributions qualitatively similar to those of Busana et al.’ 

Exact solution: we have removed this term because all solutions require a degree of numerical estimation. 

Remove Orignal Line 34 ‘exact solutions’. 

Comment 12

Abstract, 34-35. The statement is unclear. What is the purpose of the method that is failing? Performing numerical discretization of the mathematical model? Or performing data assimilation?

Response to Comment 12

The classical equations referred to here model convective transport of gases to and from well mixed compartments, and assume that diffusion can be neglected (apart from equilibration across the alveolar membrane). This assumption is reasonable under most conditions. However, the high inspired PO2 in these patients coupled with the extraordinarily wide V/Q ratios will create certain regions of the lung where convective tramsport in the gas phase is very slow, but at the same time where very large concentration gradients are predicted to persist. Under these conditions, we think the assumption that diffusion can be neglected breaks down and the model is no longer valid. 

Change Original Lines 35 - 38 ‘Finally, we note that an assumption of the classical V̇/Q̇ mass balance equations is unlikely to hold at very low V̇/Q̇ under conditions of high inspired oxygen, and therefore these distributions are an unlikely cause of the observed blood gas values in the patients.’

To R3 Lines 35 - 37 ‘We consider that these wide distributions arise because the assumed value for shunt flow is too low in these patients, and we discuss possible reasons why the assumption relating to shunt flow fraction may break down in COVID-19 pneumonia. ’

Comment 13

Abstract, 36. What is the conclusions and the proposed remedy to the unlikely assumption made on mass conservation?

Response to Comment 13

See response to comment 5. Fundamentally, we think the shunt flow assumption is not sufficiently accurate. 

Reviewer #2

Comment 1

The authors refer to the work of Busana et al in terms too competitive, providing definite statements on such a work, e.g.

“… they subsequently failed to find any accurate solutions for potential …”

“… result from the hypothesis of Busana et al. are unlikely to represent the true state of gas exchange in these severely ill COVID-19 patients.”

The authors should clearly state that their model is an extension of that of Busana et al, with an improvement in some sense of the results, acknowledging the fact that the Busana model is the starting point.

Response to comment 1

Sorry. We are glad the reviewer makes this point as it really wasn’t intended. Busana et al should be congratulated for managing to obtain these data during the course of the pandemic. We have checked the wording and changed where appropriate. We have also clarified that what we set out to achieve was an alternative approach to Busana et al’s multiple simulation approach that could provide a direct calculation of parameterisations that should reproduce a patient’s arterial PCO2 and PO2. 

Remove Original Lines 33 - 34 ‘We conclude that the absence of accurate solutions in Busana et al. arose through a failure of their method.’

Remove Original Line 388 ‘they subsequently failed to find any accurate solutions for potential V̇/Q̇ distributions’ removed. 

Change Original Lines 458 - 459 ‘In conclusion, the V̇/Q̇ distributions that result from the hypothesis of Busana et al. are unlikely to represent the true state of gas exchange in these severely ill COVID-19 patients.’ 

To R3 Lines 512 - 514 ‘In conclusion, the V̇/Q̇ distributions that arise when the shunt blood flow fraction is assumed proportional to the non-aerated lung fraction are unlikely to represent the true state of gas exchange in these severely ill COVID-19 patients.’

Change Original lines 397 - 398 ‘Why the approach of Busana et al. failed to deliver solutions very close to the patients’ arterial PCO2 and PO2 values is not entirely clear.’ 

To R3 Lines 453 - 454 ‘Why the approach of Busana et al. did not lead to solutions very close to the patients’ PaCO2 and PaO2 values is not entirely clear.’

Comment 2

Related to the previous point, the authors used too much strong statements about their results, e.g.:

“… we have demonstrated that no such distributions exist for their first …”

“… we have demonstrated that multiple, indeed infinite sets, of potential V̇/Q̇ distributions exist…”

It seems to me that the results of this paper are more reasonable than the ones found in Busana et al, but that there is no any validation, so I would avoid to use “demonstrate”.

Response to comment 2

We don’t completely agree with the reviewer here.

Busana et al found no possible solutions for their first patient. We use the three-compartment model to show that the postulated blood flow to the aerated lung is simply too low to support the gas exchange required – there is no solution to the problem (and therefore the starting assumptions concerning shunt flow have to be wrong).

For the second statement, the key word is ‘potential’ (or candidate) distributions because they do reproduce the patient’s arterial PCO2 and PO2. However, we completely agree with the reviewer re: validation and have adjusted the text accordingly. 

Comment 3

Related to the previous point: The authors used the available data for the parameter calibration, but not for the validation, is this true? If yes, this should be clearly stated in the text

Response to comment 3

Yes, that is completely correct. Furthermore, we argue that the extraordinarily-wide distributions are unlikely to exist in these patients given their very high inspired PO2. Thus, if there were some way of validating these results experimentally, we would expect the validation to fail. We think the problem resides with the ‘shunt flow assumption’, as covered in the Discussion. 

R3 Lines 481 - 483 ‘Turning from the methodology to the results, what is evident from our solutions for the V̇/Q̇ distributions from both the methods we employed (Fig. 2 and Fig. 4) is that the assumption that the shunt fraction is proportional to the fraction of non-ventilated lung (proposed by Busana et al) led to compartments with extremely low estimates for V̇/Q̇.’

Comment 4

Also the number of cases (5) is not enough to demonstrate anything. It is noticeable to have some data after 1 year from pandemic, but the authors should again change the tone of their sentences, without any definite answer. Their results in fact “seem to show that …. “

Moreover, if I well understood, the data were obtained by Busana et al and this is another reason why the authors should refer to this paper in different terms (see point 1)

Response to comment 4

Whilst we can demonstrate things as they apply to individual patients, we agree with the reviewer that 5 patients are too few from which to draw generalized conclusions in relation to the patient population as a whole. We have checked the manuscript to ensure any generalizations have that caveat clearly stated. We also agree with the second point (see our response to point 1).

Comment 5

Methods – Overview: The journal is read by scientists with different expertise, thus I suggest to better contextualize the physical processes and the method. For example: What are the compartments? What is the pure shunt and pure deadspace? Not all the readers are familiar with this.

Response to comment 5

This is a very fair comment. We have revised the manuscript to include those definitions, but also more generally, we now start the methods with a description of the compartmental model on which this study is based.

Insert R3 Lines 90 - 126 An overview section has been introduced into the methods. 

Comment 6

Eqns (1) and (2): I am not sure that the authors explicitly give the expression of g. In any case, Eqn (2) is useless, is the same of (1)

Response to comment 6

No we don’t give an explicit expression for g. In this study, we use quite a detailed mathematical model of the blood, which involves the simultaneous solution of four non-linear equations for the plasma H+ concentration, the intra-erythrocytic H+ concentration, the permeant strong ion difference in the plasma and the permeant strong ion difference in the red cells. These values are then substituted into other equations to get the blood CO2 and O2 contents. If we try to set this all out here (apart from anything else, there are 50-100 chemical constants), it will serve to confuse rather than help. It is better just to give the reference here (O’Neil at al) so that the interested reader can follow it up. Other readers only need to know what it is doing in order to follow the paper.

We agree equation 2 was probably unnecessary – the only points we were trying to make were that the inverse function exists (i.e. the function is both injective and surjective) and that it is possible to use it in calculations. 

Change Original lines 128 and 129 two equations 

To R3 line 165 one equation. 

Comment 7 

Methods: the authors should summarize all the procedure with a final algorithm or better with a flowchart, highlighting:

- the input

- how could they be obtained (measures, assumptions, …)

- the output

- their clinical relevance

Response to comment 7

This is an excellent idea and we have now introduced a flowchart into the Methods section. 

Insert R3 Figure 1 as a flowchart. 

Comment 8

Figures 1 and 2: It seems to me that different behaviours are experienced by Patients 2,3 vs 4,5. Please comment on this.

Response to comment 8 

There are indeed some quantitative differences, and of course the patients differ in their arterial blood gases and the level of inspired O2 that they are receiving (table 1). However, qualitatively the results are similar. They are all monotonic increasing functions and the solutions are all confined to exceptionally low values for V/Q. 

Comment 9 

Details on the numerical methods used to find solutions should be provided

Response to comment 9

We used the in-built equation solvers in Matlab (there is nothing particularly tricky about the solutions). We have added this point to the methods and referred to the particular solvers that were employed. 

Insert R3 Lines 114 - 115 ‘Where numerical solutions were required, Matlab’s inbuilt solvers fsolve and fminbnd were used.’

Comment 10

The clinical relevance of the results should be discussed.

Response to comment 10

We have added a section in the Discussion about the results’ clinical relevance.

Insert R3 lines 439 - 452, and Insert R3 lines 471 - 480 to discuss clinical relevance. 

Comment 11 

Limitations and future perspective should be added

Response to comment 11

A section on limitations and future perspective has been added in the Discussion. 

Insert R3 line 506 - 511 to discurss limitations and future perspectives. 

Comment 12

Line 294 should be after the caption.

Response to comment 12

Done. 

RESPONSE TO REVIEWERS (2nd REVISION)

Reviewer #1

The paper improved after a significant revision that partially incorporated the major and minor questions. I think the paper has merit, even if some major points persist and must be thoroughly addressed.

Comment 1 

I am very confused by the use of the words “solution”, “model”, and “problem”. Mathematically speaking, a solution is a value, number, function, etc… that fulfill the conditions set by equations. Here, I find very hard to understand which are the equations to be solved (models and problems) and which variables are the solutions by reading through the text. Figure 1 is adding confusion rather than giving a clear picture of the problems/equations and solutions searched for. What is meaning that a “solution is not possible”? I also find very difficult to identify data and solutions (?) in the models.

Response to comment 1

Thank you – this is very helpful comment to guide further revision. We have added a paragraph at the beginning of the Methods to define the terms ‘solution’, ‘model’ and ‘problem’.

A solution is a set of paired values for ventilation and perfusion {Vi, Qi}, that will result in calculated values for arterial PCO2 and PO2 that, to within experimental error, match the measured values from the patient. We illustrate solutions in the figures. The meaning of a ‘solution is not possible’ is that, after applying constraints relating to the patient, there are no sets {Vi, ,Qi} that form a solution to the problem. This can arise if one or more of the constraints is wrong.

Insert R3 Lines 91 - 103 to define ‘solution’, ‘model’ and ‘problem’. 

Comment 2

The concept of “distribution” should be introduced and carefully presented. If I understand correctly, this work revolves around the idea of calculating such distribution instead of a sort of “trial and error” approach by Busana et al., who explore the space of plausible parameters to identify a plausible combination of these ones. Can this distribution interpreted instead as a combination of values?

Response to comment 2

We have added a definition of distribution in the paragraph at the start of the Methods. Busana et al do not state what their parameters are, but basically their parameters are used to generate the ‘ V/Q distribution’, which is the set of paired values {Vi, Qi}. The number of paired values in this set can be varied. We explore the three-compartment model of the lung – which has three pairs of values in this set (and has a very important place in the development of theory around gas exchange in the lung), a four-compartment model of the lung, and a ‘multi’-compartment model of the lungs, where the number of Vi, Qi pairs is very much higher and where it is really being used as a computational approximation to a continuous distribution (where the pairs of values Vi, Qi in the set {Vi, Qi} would be infinite).

So the answer to your question is yes. The distribution is the set {Vi, Qi}.

Insert R3 Lines 91 - 103 to define ‘distribution’. 

Comment 3

118: “conservation of matter”. Shouldn’t be conservation of mass instead? Here, we are not working in the framework of relativistic mechanics, but of classical mechanics.

Response to comment 3

We have changed this.

Change R1 line 118 ‘conservation of matter’ 

To R3 line 133 ‘conservation of mass’. 

Comment 4: 150: Eq. (5). g is not defined, if not much later in the text. Similarly for other mathematical notation.

Response to comment 4

As indicated in the preceding paragraph, g is a function that represents the blood gas model of reference 7. To make this clearer, we have now added the reference in the sentence immediately preceding equation 5. 

Change R1 lines 148 - 149 ‘The model is represented here by the vector function:’ 

To R3 lines 163 - 164 ‘The model described in [7] is represented here by the vector function g:’.

Comment 5

Equations are numbered in round brackets (XY). However, in the text they are referred to as Eq XY. Instead, bibliographic references are cited in round brackets (in place of more common squared brackets), which is adding confusion to the “Method” section.

Response to comment 5

We have revised the manuscript so that the citations now appear in the text in squared brackets.

Comment 6

356-365: are these part of the table caption?

Response to comment 6

Yes, that is correct. 

Reviewer #2

It is hard to evaluate the changes made by the authors. They should clearly indicate in their answers the number of pages and lines where changes have been made and, possibly, indicate in colours such changes in the text.

Response to reviewer #2

Our apologies for this. We used the track changes feature to highlight all the text that had changed, but there was so much of it, and so much editing, that the end result was a bit of a mess. The changes are much clearer in this second revision.

---

## [Decision Letter · Decision Letter 3]

5 Aug 2022

Identifying putative ventilation-perfusion distributions in COVID-19 pneumonia

PONE-D-21-23433R3

Dear Dr. Robbins,

We’re pleased to inform you that your manuscript has been judged scientifically suitable for publication and will be formally accepted for publication once it meets all outstanding technical requirements.

Kind regards,

António M. Lopes, PhD

Academic Editor

PLOS ONE

Additional Editor Comments (optional):

Reviewers' comments:

Reviewer's Responses to Questions

**Comments to the Author**

1. If the authors have adequately addressed your comments raised in a previous round of review and you feel that this manuscript is now acceptable for publication, you may indicate that here to bypass the “Comments to the Author” section, enter your conflict of interest statement in the “Confidential to Editor” section, and submit your "Accept" recommendation.

Reviewer #2: All comments have been addressed

2. Is the manuscript technically sound, and do the data support the conclusions?

Reviewer #2: Yes

3. Has the statistical analysis been performed appropriately and rigorously? 

Reviewer #2: N/A

4. Have the authors made all data underlying the findings in their manuscript fully available?

Reviewer #2: Yes

5. Is the manuscript presented in an intelligible fashion and written in standard English?

Reviewer #2: Yes

6. Review Comments to the Author

Reviewer #2: The authors answered to all my issues.I suggest the paper for publication on the journal.

Best regards

7. PLOS authors have the option to publish the peer review history of their article (what does this mean?). If published, this will include your full peer review and any attached files.

Reviewer #2: No

---

## [Editor Report · Acceptance letter]

19 Aug 2022

PONE-D-21-23433R3 

Identifying putative ventilation-perfusion distributions in COVID-19 pneumonia 

Dear Dr. Robbins:

I'm pleased to inform you that your manuscript has been deemed suitable for publication in PLOS ONE. Congratulations! Your manuscript is now with our production department. 

Kind regards, 

on behalf of

Dr. António M. Lopes 

Academic Editor

PLOS ONE